# Embodied Navigation Foundation Model

**Jiazhao Zhang**[1,2,*] **Anqi Li**[1,2,*] **Yunpeng Qi**[3,4,*] **Minghan Li**[2,*] **Jiahang Liu**[2]
**Shaoan Wang**[1] **Haoran Liu**[1,2] **Gengze Zhou**[5] **Yuze Wu**[6] **Xingxing Li**[6] **Yuxin Fan**[6]
**Wenjun Li**[6] **Zhibo Chen**[3] **Fei Gao**[6,7] **Qi Wu**[5] **Zhizheng Zhang**[2,4,†] **He Wang**[1,2,4,†]

[1]Peking University  [2]Galbot  [3]USTC  [4]BAAI
[5]University of Adelaide  [6]Zhejiang University  [7]Differential Robotics
Project Page: https://pku-epic.github.io/NavFoM-Web/

## Abstract

Navigation is a fundamental capability in embodied AI, representing the intelligence required to perceive and interact within physical environments. To achieve such intelligence, recent advanced works leverage Vision-Language Models (VLMs), which demonstrate strong generalizability and possess a well-suited formulation for navigation. However, these approaches remain largely confined to narrow task settings and embodiment-specific architectures. In this work, we introduce a cross-embodiment and cross-task Navigation Foundation Model (NavFoM), trained on eight million navigation samples that encompass quadrupeds, drones, wheeled robots, and vehicles, and spanning diverse tasks such as vision-and-language navigation, object searching, target tracking, and autonomous driving. NavFoM employs a unified architecture that processes multimodal navigation inputs from varying camera configurations and navigation horizons. To accommodate diverse camera setups and temporal horizons, NavFoM incorporates identifier tokens that embed camera view information of embodiments and the temporal context of tasks. Furthermore, to meet the demands of real-world deployment, NavFoM controls all observation tokens using a dynamically adjusted sampling strategy under a limited token length budget. Extensive evaluations on seven public benchmarks demonstrate that our model achieves state-of-the-art or highly competitive performance across different navigation tasks and embodiments without requiring task-specific fine-tuning. Additional real-world experiments further confirm the strong generalizability and practical applicability of our approach.

## 1 Introduction

For both embodied agents and humans, navigation serves as a foundational capability that enables them to move intelligently within physical environments to accomplish specified tasks (Shah et al., 2023a; Bar et al., 2025; Zhang et al., 2024b). Achieving robust navigation requires a deep understanding of environmental context and task instructions, typically presented through visual and linguistic observations, which are reminiscent of Visual Language Models (VLMs). However, VLMs (Liu et al., 2023a; Yang et al., 2024a; Guo et al., 2025) have recently demonstrated remarkable zero-shot generalization in tasks such as retrieval, classification, and captioning from large-scale open-world data, without reliance on domain-specific fine-tuning. In contrast, embodied navigation (Savva et al., 2019a; Deitke et al., 2022) remains tied to narrow task domains, embodiment-specific architectures, and restricted instruction formats.

In pursuit of generalist navigation, the community has witnessed growing interest (Zhang et al., 2024a; Cheng et al., 2025; Shah et al., 2023a; Long et al., 2024), yet progress has been hindered by the constrained design and limited domain applicability of prior research. In cross-task navigation, previous methods (Zhang et al., 2025a; Yin et al., 2025; Zhu et al., 2025) typically assume a consistent camera configuration for the robot and unify various tasks such as vision-and-language navigation, object searching, and target tracking. For cross-embodiment navigation, current approaches (Eftekhar et al., 2024; Hirose et al., 2023) implicitly learn priors about the physical shape

---

[*]Joint First Author  [†] Corresponding Author

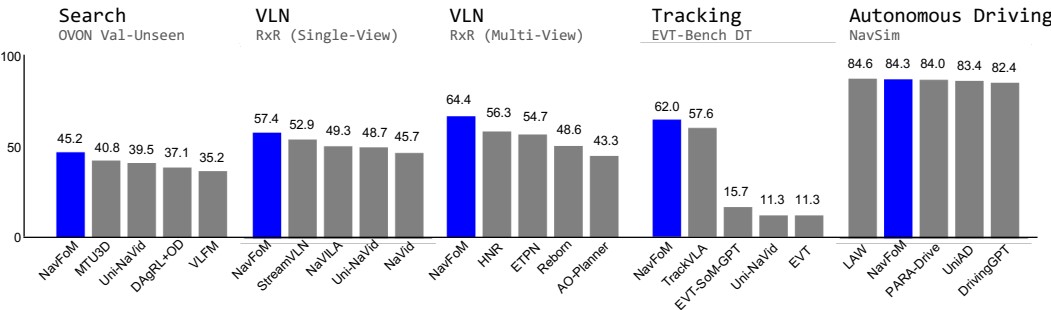

Figure 1: Benchmark performance of NavFoM, we compare our method with SOTA baselines on each benchmarks. See Section 3 for more detials.

of the embodiment but are often restricted to specific navigation tasks. The existing divergence between navigation tasks and embodiments highlights the absence of a foundational navigation model capable of handling different tasks across diverse embodiments.

In this work, we toward building a cross-task and cross-embodiment embodied navigation foundation model, NavFoM, trained on eight million navigation samples spanning diverse embodiments and tasks. Inspired by humans' ability to accomplish a wide range of navigation tasks primarily through visual sensory input and the recent success of vision-only navigation methods (Shah et al., 2023a; Zeng et al.), we formulate the generalist navigation task as processing egocentric videos (captured by one or more cameras mounted on the robot) alongside language instructions, and predicting subsequent trajectories to fulfill those instructions. This formulation is compatible with most existing navigation task settings (Contributors, 2023; Wang et al., 2024a).

To align generalizable embodiments across diverse camera configurations, we introduce temporal-viewpoint indicator tokens (TVI tokens) to identify both the viewpoint of camera setups and the temporal information of the navigation horizon. By dynamically adjusting these TVI tokens, our method enables co-tuning across different camera setups and supports joint training with both image-QA and video-QA samples (Shen et al., 2024; Li et al., 2023). Furthermore, to address the constraints of practical deployment such as hardware memory cost and inference speed, we propose a token Budget-Aware Temporal Sampling (BATS) strategy, which dynamically samples navigation history tokens based on a forgetting curve constrained by a token budget. This token sampling approach balances performance and inference speed, enhancing the practicality for real-world deployment.

We collected a comprehensive and diverse navigation dataset comprising 8.02 million samples, sourced from public navigation datasets (Savva et al., 2019a; Wang et al., 2025c; Contributors, 2023; Wang et al., 2024a) and pseudo web-video navigation data (Li et al., 2025a). The dataset includes cross-embodiment trajectories from quadruped robots, drones, wheeled robots, and cars, covering a wide range of tasks such as vision-and-language navigation, object searching, target tracking, and autonomous driving. These navigation samples feature diverse instructions and scenarios that require multiple skills, enabling NavFoM to acquire generalized navigation capabilities. Additionally, we gathered 4.76 million open-world knowledge samples (Shen et al., 2024; Li et al., 2023) derived from both image-based and video-based question-answering tasks. Following the approach of (Zhang et al., 2024a), we co-tune the navigation data together with image and video QA data in an end-to-end manner, facilitating large-scale and comprehensive training of NavFoM.

Our experiments demonstrate that NavFoM achieves substantial advancements in generalist navigation. Without task-specific fine-tuning, NavFoM attains state-of-the-art or competitive performance across diverse public benchmarks for a variety of embodiments. On VLN-CE RxR (Ku et al., 2020a), NavFoM improves performance in multi-camera settings (from 56.3% to 64.4% SR) and in single-camera settings (from 51.8% to 57.4% SR) compared to prior baselines. On HM3D-OVON (Yokoyama et al., 2024b), our method achieves 45.2% SR in a zero-shot setting, outperforming the previous fine-tuned SOTA method (43.6% SR). Similarly strong results are observed across various benchmarks in object searching, tracking, and autonomous driving. We further validate NavFoM through real-world experiments on multiple robotic platforms, including humanoid robots, quadrupeds, drones, and wheeled robots. These results underscore its strong generalizability and highlight promising progress toward generalist navigation.

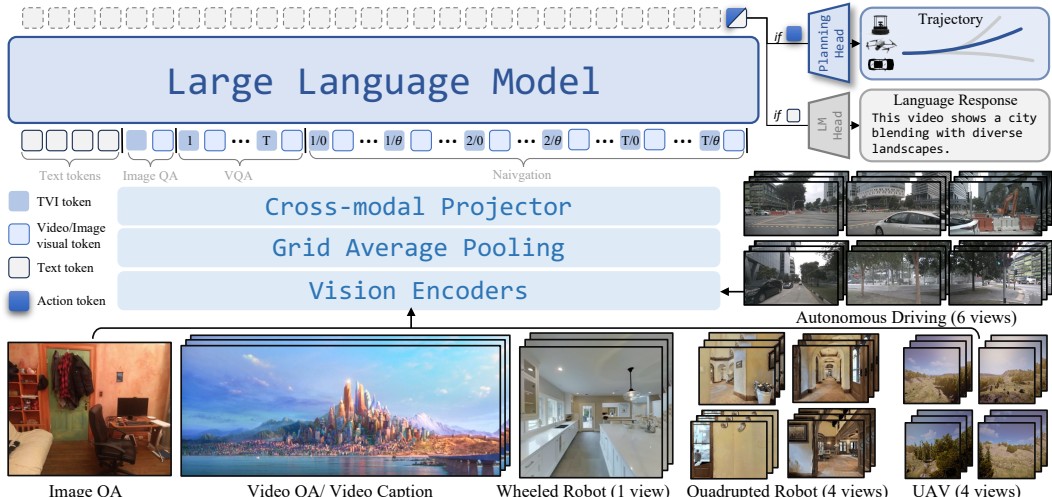

Figure 2: **Pipeline of NavFoM.** Our method provides a unified framework for handling multiple tasks, including Image QA, Video QA, and Navigation. We organize text tokens and visual tokens using temporal-viewpoint indicator tokens (Sec. 2.1.1).

## 2 METHOD

**Generalist Navigation Task.** We consider a general navigation setting in which a mobile embodiment is given a textual instruction $L$ and a sequence of images $I_{1:T}^{1:N} \in \mathbb{R}^{W \times H \times 3}$, captured on-the-fly from $N$ different cameras at time steps $\{1, ..., T\}$. Given these observations and the instruction, our model $\pi$ is required to predict a navigation trajectory $\tau = \{\mathbf{a}_1, \mathbf{a}_2, ...\}$, where each $\mathbf{a} \in \mathbb{R}^4 = (x, y, z, \theta)$ represents a position and orientation waypoint. Note that $z$ is only used when the embodiment is a UAV, and $\theta$ denotes the yaw angle (since our task does not require agile flight motions, the yaw angle suffices). The model drives the mobile embodiment to fulfill the instruction according to the mapping $\pi(L, I_{1:T}^{1:N}) \mapsto \tau_T$.

**Basic Architecture.** We extend vanilla video-based vision-language models (VLMs) (Li et al., 2023; Shen et al., 2024) to a dual-branch architecture for both navigation and question-answering (Wang et al., 2025c). For navigation, we first encode the observed images $I_{1:T}^{1:N}$ using vision encoders and a cross-modality projector (Liu et al., 2023a) to obtain visual tokens $E_{1:T}^{1:N}$. The instruction is embedded following common practices in existing language models (Liu et al., 2023a) to produce language tokens $E_L$. The visual tokens are then organized via temporal-viewpoint indicator tokens (sec. 2.1.1) and budget-aware temporal sampling (sec. 2.1.2), concatenated with the language tokens, and fed into a large language model to predict the action token. This token is subsequently decoded by a planning model to generate a waypoint-based trajectory.

$$E_T^A = \mathrm{LLM}(E_{1:T}^{1:N}, E_L),$$
$$\tau_T = \mathrm{ActionModel}(E_T^A). \tag{1}$$

For the question-answering task, we follow existing methods Liu et al. (2023a) and predict the next token in an auto-regressive manner. As in existing works (Zhang et al., 2024a; 2025a; Wang et al., 2025c; Cheng et al., 2025), our model enables the co-tuning of both navigation and QA samples.

### 2.1 NAVIGATION FOUNDATION MODEL

**Observation Encoding.** Given captured egocentric RGB sequences $I_{1:T}^{1:N} \in \mathbb{R}^{W \times H \times 3}$ from $N$ multi-camera views at time step $T$, we employ pre-trained visual encoders (DINOv2 (Oquab et al., 2023) and SigLIP (Zhai et al., 2023), a widely used recipe (Kim et al.; Tong et al., 2024)) to extract visual features $\mathbf{V}_{1:T}^{\mathrm{dino/SigLIP}} \in \mathbb{R}^{P \times C}$, where $P$ is the number of patches (set to 576) and $C$ represents the embedding dimension. For token savings and computational efficiency, we directly concatenate $V_{1:T}^{\mathrm{dino}}$ and $V_{1:T}^{\mathrm{siglip}}$ along the channel dimension and denote the resulting representation

as $V_{1:T}$. During navigation, on-the-fly captured videos leads an extensive number of frames, which subsequently produce an extensive set of visual features. To address this, we employ a grid pooling strategy (Zhang et al., 2024a; 2025a) (Figure 2, Grid Average Pooling) on the visual features to generate more compact representations. Specifically, we utilize two resolution scales:

$$\mathbf{V}^{\text{fine}/\text{coarse}} = \text{GridPool}(\mathbf{V}, \frac{64}{P} \, or \, \frac{4}{P}), \tag{2}$$

where $V^{\text{fine}} \in \mathbb{R}^{64 \times C}$ provides fine-grained observations, while $V^{\text{coarse}} \in \mathbb{R}^{4 \times C}$ offers coarse-grained observations. In this case, we use fine-grained features $V_{\text{fine}}$ for the latest navigation observation and image QA (at time step $T$), while using coarse-grained features for navigation history and video data (across time steps $1 : T$). Finally, following established VLMs (Liu et al., 2023a; Li et al., 2023), we use a cross-modality projector $\mathcal{P}(\cdot)$ (a 2-layer MLP) to project visual features into the latent space of the Large Language Model: $\mathbf{E}_T^V = \mathcal{P}(V_{1:T}^{1:N})$.

### 2.1.1 TEMPORAL-VIEWPOINT INDICATOR (TVI) TOKENS.

Given that visual tokens do not inherently incorporate viewpoint and temporal information, a key challenge in multi-view navigation models lies in enabling the LLM to discern which tokens correspond to different timesteps or distinct camera viewpoints. Previous approaches were limited to either specific camera configurations or embodiments (Long et al., 2024; Gao et al., 2025) or simply concatenated tokens from all viewpoint images (Zheng et al., 2024; Fu et al., 2025b), thereby overlooking the flexibility of LLM token organization. To enable flexible processing of arbitrary camera arrangements, we introduce temporal-viewpoint indicator tokens, inspired by the demonstrated effectiveness of specially designed tokens for time/modality/task identification (Guo et al., 2025; Chen et al., 2023), an approach that has been widely recognized to facilitate LLM learning. In our setting, the indicator tokens are used in diverse tasks, including image QA, video QA, and navigation, which should meet three important attributes:

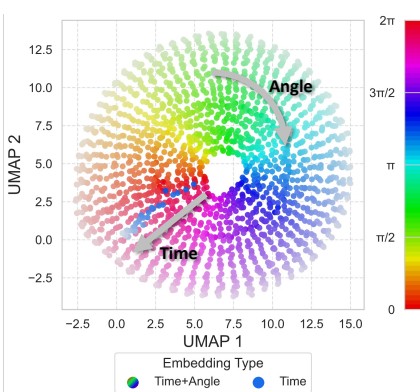

Figure 3: **Visualization of Temporal-Viewpoint Indicator (TVI) tokens.** We employ a clustering algorithm (McInnes et al., 2018) to map high-dimensional embeddings into a 2D space.

- *Viewpoint-Awareness*: The token's angle embedding must preserve the circular continuity of azimuthal angles (*e.g.*, $0 \equiv 2\pi$), ensuring that the distance metric between embeddings reflects geometric proximity (*e.g.*, $d(0, \epsilon) < d(0, \pi)$ when $\epsilon \neq \pi$).

- *Time-Awareness*: The token must uniquely identify the temporal order of frames across all camera views, while maintaining robustness to irregular sampling intervals.

- *Separability*: The indicator tokens may encode either viewpoint or temporal information (for video QA) or may exclude such information entirely (for image QA).

To meet these requirements, our Temporal-Viewpoint Indicator (TVI) tokens $\mathbf{E}_{\text{TVI}} \in \mathbb{R}^C$ (where timestep and view angle are denoted as $t$ and $\phi$, respectively) consist of three types of embeddings: angle embedding $\text{AnglePE}(\phi) \in \mathbb{R}^C$, time embedding $\text{TimePE}(t) \in \mathbb{R}^C$, and a learnable base embedding $\mathbf{E}_{\text{Base}} \in \mathbb{R}^C$:

$$\mathbf{E}_{\text{TVI}} = \begin{cases} \mathbf{E}_{\text{Base}} + \mathcal{P}_{\text{time}}(\text{TimePE}(t)) + \mathcal{P}_{\text{angle}}(\text{AnglePE}(\phi)), & \text{if Navigation} \\ \mathbf{E}_{\text{Base}} + \mathcal{P}_{\text{time}}(\text{TimePE}(t)), & \text{if Video QA} \\ \mathbf{E}_{\text{Base}}, & \text{if Image QA} \end{cases} \tag{3}$$

where $\text{AnglePE}(\phi)$ is implemented using a concatnation of sinusoidal position encodings (Vaswani et al., 2017) applied to the cosine and sine values of the azimuthal angles separately, and $\text{TimePE}(t)$ is implemented as a sinusoidal position encoding of $t$. Here, $\mathcal{P}_{\text{time}}$ and $\mathcal{P}_{\text{angle}}$ are both implemented as two-layer MLPs (similar in design to those used in Liu et al. (2023a)). For different tasks and TVI tokens, we employ different combinations of indicator token components to represent the attributes of various visual tokens. For the navigation task, we include both temporal and viewpoint

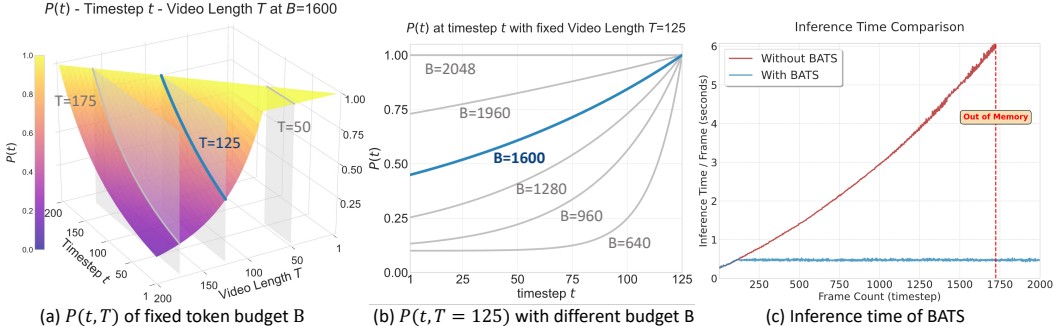

Figure 4: **Visualization of BATS and corresponding time cost.** (a) Given a fixed token budget $B = 1600$, we illustrate the sampling probability at different timesteps $t$ for the latest timestep $T$. (b) Given a maximum timestep $T = 125$, we plot the sampling probability across different timesteps $t$ under varying token budgets $B$. (c) We compare the inference time when using BATS versus not using BATS (keeping all frames).

information. For the video QA task, we incorporate temporal information. For the image QA task, we use only $E_{\text{Base}}$ as an indicator that the subsequent tokens are visual tokens. This strategy offers a flexible approach to organizing significantly different sample types and facilitates LLM learning (Sec. 2.1.3). We provide a plot of the clustering results (McInnes et al., 2018) of TVI Tokens in Figure 3, where we observe that the tokens are distinguished from one another according to the viewpoint $\theta$ (represented by a rainbow colorbar) and the timestep $t$ (represented by color value).

### 2.1.2 BUDGET-AWARE TEMPORAL SAMPLING (BATS).

During navigation, on-the-fly captured video can generate an excessive number of visual tokens, increasing both inference and training time and hindering real-world deployment. Previous methods address this challenge in two ways: (1) Token Merging (Zhang et al., 2025a), which introduces additional computational overhead during training and leads to inconsistent inference speeds during evaluation; (2) Uniform Sampling (Cheng et al., 2025), which often fails to adequately capture recent observations due to a lack of short-term context. Moreover, in scenarios involving variable camera-view settings (where the number of frames increases significantly) both strategies require additional modifications.

To this end, we propose *Budget-Aware Temporal Sampling (BATS)*, which is designed for (a) practical purposes (i.e., constraining the maximum token length to accommodate inference speed and GPU memory limitations), (b) retaining more recent information to enhance understanding and planning while preserving sufficient historical context for navigation, and (c) direct adaptability to varying numbers of cameras. Specifically, given a token budget $B_{\text{token}}$ and a multi-view video sequence $I_{1:T}^{1:N} \in \mathbb{R}^{W \times H \times 3}$, we employ an exponential growth based sampling probability $P(t)$, which is inspired by the "forgetting curve". In this case, when the number of captured frame tokens exceeds the token budget, we compute a sampling probability for each frame:

$$P(t) = (1 - \epsilon)e^{k(t-T)/T} + \epsilon, \quad k > 0, \tag{4}$$

where the $\epsilon$ (we use $\epsilon = 0.1$) ensures that the lower bound of sampling probability is in the approximate range and the $k$ denotes the exponential decay rate. Therefore the expected number of sampled frames can be computed as:

$$\mathbb{E}_{\text{frames}} \approx \int_0^T P(t)dt = (1 - \epsilon)\frac{1 - e^{-k}}{k}T + \epsilon T \tag{5}$$

We constrain the expected number of tokens $((4+1)\mathbb{E}_{\text{frame}} + (64+1))N$ to be no larger than $B_{\text{token}}$. This implies $\mathbb{E}_{\text{frame}} \leq \frac{B_{\text{token}} - (64+1)N}{(4+1)N}$, and with sufficiently large number of frames $T$, the number of sampled frames will converge to the expectation (Figure 4 (c)). We can offline calculate $k$ for different $T$ using Brent's method (Brent, 2013), leading correspongding $P(t)$ (Equation 4). Note that since we set the lower-bound probability $\epsilon$, Equation 5 may become unsolvable for very large $T$ (*e.g.*, $T = 1120$ under a four-camera setup with a token budget $B_{\text{token}} = 2048$). However, this

situation rarely occurs (for the list task in Figure 1), as most timesteps are approximately 122 steps in VLN-CE RxR (Ku et al., 2020a). We provide the details of using BATS in Appendix A.2 and a break-in analysis of BATS in Figure 4.

### 2.1.3 LLM FOWARDING AND TRAINING DETAILS

**Token Organization.** After obtaining the visual tokens $E_{1:T}^{1:N}$ (sampled via BATS, Sec. 2.1.2) and the language tokens $E_L$, we organize these tokens using TVI Tokens (Sec. 2.1.1) for forwarding through the LLM. For navigation, we use $\mathbf{E}_{\text{Base}} + \mathcal{P}_{\text{time}}(\text{TimePE}(t)) + \mathcal{P}_{\text{angle}}(\text{AnglePE}(\phi))$ to represent both temporal and viewpoint information. Here, fine-grained visual tokens are used for the most recent observations, while coarse-grained tokens are utilized for historical observations. Our token organization strategy enhances the LLM's understanding of the input tokens and supports a unified framework for Image QA, Video QA, and navigation tasks. Further details of token orgainzation on Image QA and Video QA can be found in Appendix A.6.

**Trajectory prediction.** For the navigation task, given the predicted action hidden state $E_T^A$ from the forward pass of the LLM, we apply a plannning model $\mathcal{A}_\theta$ (implemented as a three-layer MLP) to extract the trajectory information $\tau_T$. Note that the original trajectory may range from a few meters (indoor navigation) to tens of meters (autonomous driving and drones). In this case, directly predicting the raw trajectory could lead to divergence in the waypoint distribution. Therefore, following previous methods Shah et al. (2023a), we normalize the waypoints of trajectories to a distribution of $[-1, 1]$ using a task-specific scaling factor $\alpha_{\text{task}}$. Here, we use three different scaling factors for indoor navigation, UAVs, and cars, as shown in Appendix A.1. We can formulate the trajectory prediction as follows:

$$\tau_T = \{\mathbf{a}_1, ..., \mathbf{a}_M\}_T = \alpha_{\text{task}} \cdot \mathcal{A}_\theta(E_T^A), \tag{6}$$

where $M$ is set to 8, and the normalized trajectory is rescaled to absolute values by multiplying by $\alpha_{\text{task}}$. The trajectory loss is computed using the mean squared error (MSE) $L_{\text{nav}} = \text{MSE}(\tau^{\text{idx}}, \tau_{\text{gt}}^{\text{idx}})$, there idx denotes the valid action indices. For wheeled robots/car embodiments, $\mathbf{a}^{\text{idx}} = (x, y, \theta)$; for UAVs, $\mathbf{a}^{\text{idx}} = (x, y, z, \theta)$. For the question-answering task, we employ the cross-entropy loss $L_{\text{QA}}$ under a next-token-prediction supervision framework. Given a batch containing both navigation and QA samples, the total loss is defined as $L = \beta L_{\text{nav}} + L_{\text{QA}}$. Here, $\beta$ is a constant scaling factor (set to 10) used to amplify the navigation loss, which tends to be numerically small since it is derived from mean squared error. Note that, $\beta$ is important when the training scale is small, where a large $\beta$ can accelerate convergence. We also believe a more adaptive way to adjust $\beta$ may be a promising direction for future work.

**Training Configurations.** Our model is trained on a cluster server equipped with 56 NVIDIA H100 GPUs for approximately 72 hours, resulting in a total of 4,032 GPU hours. For question-answering data, all frames are sampled at 1 FPS to reduce redundancy between consecutive frames. For discrete navigation data (e.g., Habitat environments Savva et al. (2019a)), we sample each step after the robot performs a discrete action (See Appendix A.1 for details on how discrete actions are modified into trajectories.). For continuous navigation environments (e.g., EVT-Bench Wang et al. (2025c), autonomous driving (Caesar et al., 2020b; Contributors, 2023)), data are sampled at 2 FPS to avoid redundancy. During training, the vision encoders (DINOv2 Oquab et al. (2023) and SigLIP Zhai et al. (2023)) and the large language model (Qwen2-7B Yang et al. (2024a)) are initialized with their default pretrained weights. Following the training paradigm of VLM (Liu et al., 2023a), we fine-tune only the designated trainable parameters for a single epoch.

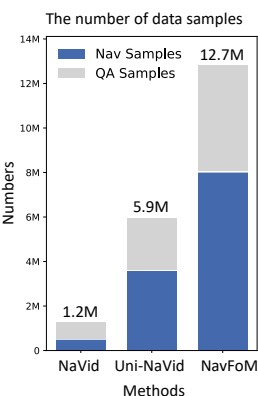

Figure 5: Comprasion of number of training samples with previouse methods.

## 2.2 DATA

To fine-tune NavFoM, we collect and process a large set of comprehensive and diverse training samples, totaling 12.7 million instances. These include 8.02 million navigation samples, 3.15 million image-based question-answering samples, and 1.61 million video-based question-answering samples. The navigation samples are collected

and processed from diverse datasets. Specifically, we collect Vision-and-Language Navigation samples (3.37 M) from R2R (Krantz et al., 2020), RxR (Ku et al., 2020a) and OpenUAV (Wang et al., 2024a); Object Goal Navigation (1.02 M) from HM3D ObjectNav (Savva et al., 2019a); Active Visual Tracking (897 K) from EVT-Bench (Wang et al., 2025c); Autonoums Driving (681 K) from nuScense (Caesar et al., 2020a) OpenScene (Contributors, 2023); and web-video navigation from Sekai dataset (Li et al., 2025a). All navigation data are collected in a unified manner, including previously captured videos (single or multiple cameras), instructions, and predicted trajectory waypoints. Further details regarding the navigation samples please refer to Appendix A.4.

Besides navigation data, we, we gather image-based QA (3.15 M) and video-based QA (1.61 M) data from off-the-shelf datasets following existing video-based Vision-Language Models (VLMs) (Shen et al., 2024; Li et al., 2023; KunChang Li & Qiao, 2023; Li et al., 2024a).

## 3 EXPERIMENTS

### 3.1 EXPERIMENT SETUP

To evaluate the performance of NavFoM, we conduct extensive experiments and ablation studies addressing three key aspects: (1) How does NavFoM perform on diverse navigation tasks across different benchmarks? (2) How well does NavFoM perform in real-world environments? (3) Are the key design components of our method effective? Our method is compared against strong baselines on each benchmark.

**Benchmarks and metrics**. We evaluate our method on various navigation tasks, including VLN (VLN-CE R2R (Krantz et al., 2020), RxR (Ku et al., 2020a) and OpenUAV (Wang et al., 2024a), searching (Yokoyama et al., 2024b), tracking (Wang et al., 2025c), and autonomous driving (Caesar et al., 2020a; Dauner et al., 2024b), which are across different embodiments (e.g., egocentric, four-camera, six-camera, and eight-camera configurations). NavFoM uses only online-captured egocentric video (some from multi-view sources) and an instruction as input to predict the trajectory for the robot to execute. We adopt common evaluation metrics from these benchmarks, including success rate (SR), oracle success rate (OS), success weighted by path length (SPL), normalized Dynamic Time Warping (nDTW), navigation error from goal (NE), and the PDM score Dauner et al. (2024a). For a detailed introduction to each benchmark and metric, please refer to Appendix B.1.

**Deployment on synthetic and real-world environemnts.** For each navigation task, we adhere to the default settings established in prior works (Krantz et al., 2020; Savva et al., 2019b; Das et al., 2018; Islam et al., 2019). For simulators, we use 2048 token gudeget, a similar length as baselines Cheng et al. (2025). Note that for certain benchmarks in Habitat-Lab continuous environments that use discrete actions (such as FORWARD, LEFT, RIGHT, and STOP), we replace these discrete actions with trajectory-based actions. For real-world deployment, we employ a remote server equipped with an NVIDIA RTX 4090 GPU (use 1600 token budget) to run NavFoM. Under this configuration, the system requires approximately 19.1 GB of GPU memory and achieves a inference rate of 5 Hz (about 218 ms per trajectory prediction). Further details on deployment costs are provided in Appendix E.

### 3.2 BENCHMARK RESULTS

**VLN: Performance on VLN-CE** (Krantz et al., 2020; Ku et al., 2020a). We begin by evaluating our method on the most widely used vision-and-language instruction benchmarks—VLN-CE R2R and VLN-CE RxR—with the results presented in Table 1. We report performance under both single-camera and four-camera settings (360° observations). Note that our model is not fine-tuned on any specific camera configuration; instead, visual tokens are directly organized using temporal-viewpoint indicator tokens (Figure 10). Our method achieves SOTA performance on both benchmarks across different camera settings. Under the most challenging condition (single-view VLN-CE RxR), our method improves success rate (SR) from 51.8% to 57.4%. Notably, in multi-camera setups, our approach uses only four RGB cameras and attains an SR of 64.4%, outperforming previous SOTA methods (56.3% SR) that rely on RGB-D cameras and odometry information. We also observe a significant performance gain when transitioning from single-view to multi-view settings: an increase of 5.5% on R2R-CE and 7.0% on RxR-CE, respectively. This suggests that multi-view navigation foundation models represent a promising direction for future research. Besides success

Table 1: **Comparison on VLN-CE in Single-View and Multi-View Settings.** Here, S.RGB and M.RGB denote single-view and multi-view configurations, respectively. The symbol * indicates methods that utilize the waypoint predictor from (Hong et al., 2022).

| Method | Observation | | | | R2R Val-Unseen | | | | RxR Val-Unseen | | | |
|---|---|---|---|---|---|---|---|---|---|---|---|---|
| | S.RGB | M.RGB | Depth | Odo. | NE ↓ | OS ↑ | SR ↑ | SPL ↑ | NE ↓ | SR ↑ | SPL ↑ | nDTW ↑ |
| AG-CMTP (Chen et al., 2021a) | | ✓ | ✓ | ✓ | 7.90 | 39.0 | 23.0 | 19.0 | - | - | - | - |
| R2R-CMTP (Chen et al., 2021a) | | ✓ | ✓ | ✓ | 7.90 | 38.0 | 26.0 | 22.0 | - | - | - | - |
| HPN+DN* (Krantz et al., 2021) | | ✓ | ✓ | ✓ | 6.31 | 40.0 | 36.0 | 34.0 | - | - | - | - |
| CMA* (Hong et al., 2022) | | ✓ | ✓ | ✓ | 6.20 | 52.0 | 41.0 | 36.0 | 8.76 | 26.5 | 22.1 | 47.0 |
| VLN○BERT* (Hong et al., 2022) | | ✓ | ✓ | ✓ | 5.74 | 53.0 | 44.0 | 39.0 | 8.98 | 27.0 | 22.6 | 46.7 |
| Sim2Sim* (Krantz & Lee, 2022) | | ✓ | ✓ | ✓ | 6.07 | 52.0 | 43.0 | 36.0 | - | - | - | - |
| AO-Planner (Chen et al., 2024a) | | ✓ | ✓ | | 5.55 | 59.0 | 47.0 | 33.0 | 7.06 | 43.3 | 30.5 | 50.1 |
| GridMM* (Wang et al., 2023b) | | ✓ | ✓ | ✓ | 5.11 | 61.0 | 49.0 | 41.0 | - | - | - | - |
| Ego²-Map* (Hong et al., 2023) | | ✓ | ✓ | ✓ | 5.54 | 56.0 | 47.0 | 41.0 | - | - | - | - |
| DreamWalker* (Wang et al., 2023a) | | ✓ | ✓ | ✓ | 5.53 | 59.0 | 49.0 | 44.0 | - | - | - | - |
| Reborn* (An et al., 2022) | | ✓ | ✓ | ✓ | 5.40 | 57.0 | 50.0 | 46.0 | 5.98 | 48.6 | 42.0 | 63.3 |
| ETPNav* (An et al., 2024) | | ✓ | ✓ | ✓ | 4.71 | 65.0 | 57.0 | 49.0 | 5.64 | 54.7 | 44.8 | 61.9 |
| HNR* (Wang et al., 2024b) | | ✓ | ✓ | ✓ | **4.42** | 67.0 | 61.0 | 51.0 | 5.50 | 56.3 | 46.7 | 63.5 |
| BEVBert* (An et al., 2023) | | ✓ | ✓ | ✓ | 4.57 | 67.0 | 59.0 | 50.0 | - | - | - | - |
| HAMT+ScaleVLN* (Wang et al., 2023c) | | ✓ | ✓ | ✓ | 4.80 | - | 55.0 | 51.0 | - | - | - | - |
| **NavFoM (Four views)** | | ✓ | | | 4.61 | **72.1** | **61.7** | **55.3** | **4.74** | **64.4** | **56.2** | **65.8** |
| LAW (Raychaudhuri et al., 2021) | ✓ | | ✓ | ✓ | 6.83 | 44.0 | 35.0 | 31.0 | 10.90 | 8.0 | 8.0 | 38.0 |
| CM2 (Georgakis et al., 2022) | ✓ | | ✓ | ✓ | 7.02 | 41.0 | 34.0 | 27.0 | - | - | - | - |
| WS-MGMap (Chen et al., 2022) | ✓ | | ✓ | ✓ | 6.28 | 47.0 | 38.0 | 34.0 | - | - | - | - |
| Seq2Seq (Krantz et al., 2020) | ✓ | | ✓ | | 7.77 | 37.0 | 25.0 | 22.0 | 12.10 | 13.9 | 11.9 | 30.8 |
| CMA (Krantz et al., 2020) | ✓ | | ✓ | | 7.37 | 40.0 | 32.0 | 30.0 | - | - | - | - |
| RGB-Seq2Seq (Krantz et al., 2020) | ✓ | | | | 10.10 | 8.0 | 0.0 | 0.0 | - | - | - | - |
| RGB-CMA (Krantz et al., 2020) | ✓ | | | | 9.55 | 10.0 | 5.0 | 4.0 | - | - | - | - |
| NaVid (Zhang et al., 2024a) | ✓ | | | | 5.72 | 49.2 | 41.9 | 36.5 | 5.72 | 45.7 | 38.2 | - |
| Uni-NaVid (Zhang et al., 2025a) | ✓ | | | | 5.58 | 53.3 | 47.0 | 42.7 | 6.24 | 48.7 | 40.9 | - |
| NaVILA (Cheng et al., 2025) | ✓ | | | | 5.22 | 62.5 | 54.0 | 49.0 | 6.77 | 49.3 | 44.0 | 58.8 |
| StreamVLN-RGB-only (Wei et al., 2025) | ✓ | | | | 5.10 | 64.0 | 55.7 | 50.9 | 6.16 | 51.8 | 45.0 | **62.1** |
| **NavFoM (Single view)** | ✓ | | | | **5.01** | **64.9** | **56.2** | **51.2** | **5.51** | **57.4** | **49.4** | 60.2 |

Table 2: **Object goal navigation.** Comparison on HM3D-OVON (Yokoyama et al., 2024b). * : denotes zero-shot evaluation. We report the performance of our method on egocentric and four-view settings.

| Method | VAL SEEN | | VAL SEEN SYNONYMS | | VAL UNSEEN | |
|---|---|---|---|---|---|---|
| | SR↑ | SPL↑ | SR↑ | SPL↑ | SR↑ | SPL↑ |
| BC | 11.1 | 4.5 | 9.9 | 3.8 | 5.4 | 1.9 |
| DAgger | 11.1 | 4.5 | 9.9 | 3.8 | 5.4 | 1.9 |
| RL | 18.1 | 9.4 | 15.0 | 7.4 | 10.2 | 4.7 |
| BCRL | 39.2 | 18.7 | 27.8 | 11.7 | 18.6 | 7.5 |
| DAgRL | 41.3 | 21.2 | 29.4 | 14.4 | 18.3 | 7.9 |
| VLFM* (Yokoyama et al., 2024a) | 35.2 | 18.6 | 32.4 | 17.3 | 35.2 | 19.6 |
| DAgRL+OD (Yokoyama et al., 2024b) | 38.5 | 21.1 | 39.0 | 21.4 | 37.1 | 19.8 |
| Uni-NaVid* (Zhang et al., 2025a) | 41.3 | 21.1 | 43.9 | 21.8 | 39.5 | 19.8 |
| MTU3D (Zhu et al., 2025) | **55.0** | 23.6 | 45.0 | 14.7 | 40.8 | 12.1 |
| **NavFoM * (Single view)** | 37.7 | 25.5 | 43.3 | 29.9 | 43.6 | 31.3 |
| **NavFoM * (Four views)** | 40.1 | **27.1** | 45.4 | **32.6** | **45.2** | **31.9** |

Table 3: **Performance on EVT-Bench**. †: Uses GroundingDINO (Liu et al., 2023b) as the open-vocabulary detector. ‡: Uses SoM (Yang et al., 2023)+GPT-4o (OpenAI, 2024) as the visual foundation model.

| Method | Single Target | | Distracted Target | |
|---|---|---|---|---|
| | SR↑ | TR↑ | SR↑ | TR↑ |
| IBVS† (Gupta et al., 2016) | 42.9 | 56.2 | 10.6 | 28.4 |
| PoliFormer† (Zeng et al.) | 4.67 | 15.5 | 2.62 | 13.2 |
| EVT (Zhong et al., 2024) | 24.4 | 39.1 | 3.23 | 11.2 |
| EVT‡ (Zhong et al., 2024) | 32.5 | 49.9 | 15.7 | 35.7 |
| Uni-NaVid (Zhang et al., 2025a) | 25.7 | 39.5 | 11.3 | 27.4 |
| TrackVLA (Wang et al., 2025c) | 85.1 | 78.6 | 57.6 | 63.2 |
| **NavFoM (Single view)** | 85.0 | 80.5 | 61.4 | 68.2 |
| **NavFoM (Four views)** | **88.4** | **80.7** | **62.0** | 67.9 |

rate, we also observe that our method achieves higher efficiency, demonstrates a higher SPL (56.2% SPL), and produces navigation trajectories that are better aligned with the instructions (65.8 nDTW).

**Perfomrence on Searching, Tracking and Autonomous Driving.**

We conduct experiments to evaluate our method across different navigation capabilities, including object goal navigation (Yokoyama et al., 2024b) in Table 2, active visual tracking (Wang et al., 2025c) in Table 3, and autonomous driving (Dauner et al., 2024a) in Table 4. We find that our approach demonstrates strong performance compared to strong baselines that are specifically designed for individual navigation tasks. Moreover, our method improves consistently when switching from a single-camera to a four-camera setup, even though it was not trained on the four-camera configuration in object navigation and tracking tasks. Additional quantitative re-

Table 4: NAVSIM `navtest` split with closed-loop metrics.

| Method | Camera | VLM-Based | PDMS ↑ |
|---|---|---|---|
| Human | - | - | 94.8 |
| Constant Velocity | - | - | 21.6 |
| Ego Status MLP | - | - | 65.6 |
| UniAD (Hu et al., 2023) | ✓ | - | 83.4 |
| PARA-Drive (Weng et al., 2024) | ✓ | - | 84.0 |
| LAW (Li et al., 2024b) | ✓ | - | **84.6** |
| DrivingGPT (Chen et al., 2024c) | ✓ | ✓ | 82.4 |
| **NavFoM (Eight views)** | ✓ | ✓ | 84.3 |

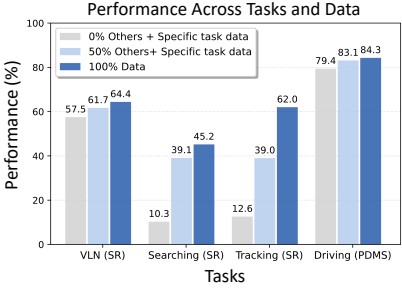

Figure 6: Visualization of real-world experiments on cross-task and cross-embodiment settings.

Figure 7: **Ablation study on the training of multiple navigation tasks.** We report the performance of different training data combinations (specific task data only, specific task data with 50% other data, and specific task data with 100% other data).

| Type | RxR Val-Unseen | | | |
|---|---|---|---|---|
| | NE ↓ | SR ↑ | SPL ↑ | nDTW ↑ |
| $B = 1024$, Uniform Sampling* | 5.33 | 59.7 | 49.6 | 57.9 |
| $B = 1024$, Linear Probability Sampling | 5.28 | 61.2 | 50.9 | 58.9 |
| $B = 1024$, Budget-Aware Temporal Sampling | 4.98 | 62.5 | 53.9 | 64.1 |
| $B = 2048$, Token Merging (Zhang et al., 2025a) | 5.01 | 63.2 | 54.9 | 64.4 |
| $B = 2048$, Uniform Sampling* | 4.90 | 62.4 | 54.0 | 63.9 |
| $B = 2048$, Linear Probability Sampling | 4.89 | 63.0 | 54.6 | 64.8 |
| $B = 2048$, Budget-Aware Temporal Sampling | **4.74** | **64.4** | **56.2** | **65.8** |
| Viewpoint-history postional embedding† | 6.27 | 52.3 | 46.3 | 58.7 |
| Individual Learned Special Toekns | 5.52 | 59.1 | 52.0 | 59.6 |
| Handcraft Toekns (Equ. 3 w.o $\mathcal{P}_{angle/time}$) | 6.06 | 53.6 | 46.1 | 58.0 |
| Temporal-Viewpoint Indicator Tokens (Equ. 3) | **4.74** | **64.4** | **56.2** | **65.8** |

Figure 8: **Ablation Study on History Token Organization Strategies and Identity Tokens.** Uniform sampling is adopted from (Cheng et al., 2025). †Positional embeddings is adopted from HAMT (Chen et al., 2021b).

sults, analyses, and visual examples are provided in Appendix C, Figure 6 and Figure 13. The analysis of both benchmark and real-world experiment failure cases can be found in Appendix F.

## 3.3 ABLATION STUDY

**Synergy of training on multiple navigation tasks.** We investigate the synergistic effects of multi-navigation task training by comparing the performance of single-task training with co-tuning that incorporates additional data from other navigation tasks (Ku et al., 2020b; Yokoyama et al., 2024b; Wang et al., 2025c; Dauner et al., 2024a). We observe that co-tuning with data from diverse navigation tasks leads to consistent performance improvements across all tasks (from 50% to 100% data ratios). Notably, Searching (improving from 10.3% to 45.2%) and Tracking (improving from 12.6% to 62.0%) exhibit the most significant gains. We attribute these improvements to the discrepancy between their training conditions (primarily single-view and closed-set target categories) and the evaluation settings, which are multi-view and open-vocabulary. These results suggest that training across multiple navigation tasks helps mitigate overfitting to task-specific navigation patterns.

**Effectiveness of BATS and TVI tokens.** We conduct ablation studies to evaluate the effectiveness of our key designs, including the history token organization strategy and visual-temporal history modeling. The experiments are conducted on the VLN-CE RxR four-camera setting, and the results are presented in Table 8. We test different token strategies under different token budgets (1024 or 2048) and find that BATS outperforms other strategies in both settings, on both token budgets. Specifically, when the token budget is reduced from 2048 to 1024, BATS demonstrates a smaller performance drop (only 1.4% ↓) on the nDTW metric compared to the baselines (6.0% ↓ and 5.2% ↓). Furthermore, we compare TVI tokens with other common alternatives and find that TVI tokens achieve significantly better performance. As illustrated in Figure 3, we attribute this improvement to the well-learned temporal and viewpoint information. Moreover, compared to the common history-viewpoint positional embedding method (Chen et al., 2021b), we observe a noticeable performance drop. We believe this is due to the additional embedding components introduced for visual tokens, which may increase model complexity, while TVI provides separate information to facilitate LLM understanding. These results demonstrate the effectiveness of TVI tokens.

## 4 RELATED WORKS

There is a large body of literature (Savva et al., 2019a; Zhang et al., 2024b) on navigation across different tasks and embodiments; here we review those most relevant to our work. In cross-task navigation, recent efforts (Wang et al., 2022; Long et al., 2024; Song et al., 2025; Zhang et al., 2025a; Gao et al., 2025; Yin et al., 2025; Ruan et al., 2025) have shown that integrating data from different categories of navigation tasks can lead to stronger performance across various scenarios. For cross-embodiment navigation, prior studies (Shah et al., 2023a;b; Yang et al., 2024b; Wang et al., 2020; Eftekhar et al., 2024; Hirose et al., 2023; Putta et al., 2024; Curtis et al., 2024; Wang et al., 2025a; Zhang et al., 2025b; Geng et al., 2025) have demonstrated the potential of transformer-based policies trained on large-scale, cross-embodiment datasets to achieve robust performance across various robotic platforms. In this work, our method presents an early attempt to unify cross-task and cross-embodiment navigation within a VLA model under a unified training and evaluation framework, demonstrating strong performance in both synthetic and real-world environments.

## 5 DISSCUSION AND CONCLUSION

In this work, we propose NavFoM, which aims to push the boundaries of navigation and explore the intelligence learned from cross-embodiment and cross-task navigation data. We introduce temporal-viewpoint indicator tokens to enhance the LLM's understanding of varying camera configurations and different horizons in navigation tasks, while also enabling co-training with navigation and question-answering data. Furthermore, we employ a token budget-aware temporal sampling strategy to balance navigation performance and efficiency, facilitating a unified approach to token sampling across diverse camera setups and task horizons. Extensive experiments on both public benchmarks and real-world environments demonstrate the strong perfomrence and generability of NavFoM. We believe that NavFoM serves as a starting point toward a navigation foundation model and will attract greater attention to intelligence-centric navigation

### ETHICS STATEMENT

This work presents a generalist navigation foundation model designed to enhance the capabilities of embodied agents across diverse environments and embodiments. We acknowledge the potential societal benefits of such technology, including improved assistive robotics, search-and-rescue operations, and autonomous systems. However, we also recognize the risks associated with deploying AI-powered navigation systems in real-world settings, such as safety hazards, privacy concerns arising from visual data collection, and potential misuse. All training data were sourced from publicly available datasets, with due consideration given to ethical guidelines. The development and evaluation of our method involved rigorous real-world testing, transparency regarding its capabilities and limitations, and adherence to applicable regulations and safety standards.

### REPRODUCIBILITY STATEMENT

We provide full implementation details (Section 2), including the model architecture, training configurations, data processing procedures, and the real-world deployment framework. All datasets (Section A.4) used are publicly accessible, and hyperparameters are clearly specified in both the main paper and the appendix. The base models (Section 2.1.3), including large language models and vision encoders, are explicitly mentioned in the paper, along with a detailed training strategy. We also include specifics regarding evaluations as well as instructions for deployment in synthetic (Section 3.1) or real-world environments (Section B.4). The code, together with pre-trained model weights, will be made publicly available upon acceptance.

### ACKNOWLEDGEMENTS

We thank all the reviewers for their constructive comments and suggestions. We also thank Jianmin Wang for his help with the hardware setup. This work is supported by GALBOT.

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

## USAGE OF LLM STATEMENT

Large Language Models (LLMs) are utilized solely to enhance the quality of written content by assisting with polishing text and correcting grammatical errors.

## A IMPLEMETATION DETIALS

### A.1 ACTION PLANNING MODEL

Due to the fact that different embodiments could have distinct trajectory scales. For instance, indoor robots often move on the scale of meters while cars move on the scale of dozens of meters. We normalize the predicted trajectory scaling across different embodiments to the range [-1,1] of all dimesions by multipy a scaling factor $\alpha_{\text{task}}$, as reported in Table 5. Note that the scaling factor is not derived from the absolute maximum value of each dimension; instead, we use the 99th percentile of each dimension to avoid the influence of outlier data.

| Embodiements | x(m) | y(m) | z(m) | $\theta$(rad) |
|---|---|---|---|---|
| Indoor robots* | 1.0 | 0.433 | - | 2.09 |
| UAV* | 7.93 | 3.19 | 7.85 | 1.04 |
| Cars* | 50.8 | 14.9 | - | 1.52 |

Table 5: Scaling factors of different dimesiong of predicted tracjtort of different embodiements.

### A.2 DETIALS OF USING BATS

During navigation, initially when the number of visual tokens is within the token budget $B$, we retain all visual tokens. Once the visual tokens exceed the budget $B$, we employ BATS to sample tokens based on a forgetting curve (Sec. 2.1.2). In practice, we precompute $P(t, T)$ for a given token budget $B$ to accelerate this process. If the navigation task involves an exceptionally long horizon, such as thousands of steps (which rarely occurs), even using the minimum sampling probability $\epsilon$ may result in the visual tokens exceeding the token budget. In such cases, we simply remove the oldest frames.

### A.3 DETIALS OF FIGURE 3

We performed clustering (McInnes et al., 2018) directly on the end-to-end learned TVI tokens (Eq. 3) and visualized the embeddings using a color map based on viewpoint angle and time step. Specifically, for the navigation task (Eq. 3 row 1), we sampled 1,800 TVI token embeddings from combinations of 60 angles (distributed over $[0, 2\pi]$) and 30 time steps (ranging from 0 to 150). For the VQA task (Eq. 3 row 2), we sampled embeddings from 30 time steps ranging from 0 to 150.

### A.4 DATA PREPARATION

**Vision-and-Language Navigation** (3.37 M) requires an agent to interpret natural language instructions and egocentric visual observations, align the instructions with visual inputs, and plan subsequent actions to reach described landmarks. Following a broad definition of VLN (Zheng et al., 2024; Wang et al., 2025b; Zhou et al., 2024), we consider both indoor environments (e.g., VLN-CE on R2R (Krantz et al., 2020) and RxR (Ku et al., 2020a)) and outdoor environments (e.g., Open-UAV (Wang et al., 2024a)), deployed on robots and unmanned aerial vehicles (UAVs), respectively. For VLN-CE on R2R and RxR (2.94 M), we capture multi-view RGB videos, annotated instructions, and trajectory data while the robot follows the ground-truth path. The multi-view RGB setup consists of a fixed front-view camera and randomly sampled surrounding cameras (ranging from one to eight). Camera heights are randomized between 0.6 m and 1.5 m, and the horizontal fields of view (HFoV) vary between 75° and 120°. For the OpenUAV dataset (429 K), we record camera streams from the front, left, right, and rear views for all sequences. Other randomization strategies remain consistent with those used in the VLN-CE tasks.

**Object Goal Navigation** (1.02 M) requires a robot to explore an unseen environment and identify a described target. For the object goal navigation dataset, we follow the method of (Zhang et al., 2025a) by collecting successful episodes from L3MVN (Yu et al., 2023), a heuristic-designed ap-

proach that explicitly models the exploration and identification stages. Our data are collected from HM3D ObjectNav (Savva et al., 2019a) episodes, which require the agent to locate objects from a predefined category set (e.g., *sofa*, *chair*, and *bed*). Nevertheless, experiments show that our method generalizes to state-of-the-art open-vocabulary object goal searching, as presented in Table 7. Note that we do not employ multiple cameras or camera randomization, as we aim to maintain the same visual observation configuration as L3MVN.

**Active Visual Tracking** (897K) (Islam et al., 2019; Francis et al., 2023; Wang et al., 2025c) requires the robot to distinguish the target within dynamic and crowded environments. The target is specified via textual instructions, e.g., "Follow the man in the blue t-shirt." The agent must recognize the appearance of the human, follow the correct person according to the instructions, and maintain an appropriate distance while avoiding obstacles. For this task, we use data from EVT-Bench, consistent with (Wang et al., 2025c), which involves diverse indoor environments and hundreds of avatars with corresponding descriptions. We also incorporate camera randomization, as described in our VLN data collection process.

**Autonoums Driving** (681K) (Hu et al., 2023; Liao et al., 2024b) requires an agent to generate a safe, comfortable, and kinematically feasible trajectory for navigating complex and dynamic real-world environments. This task evaluates the agent's ability to continuously perceive its surroundings, anticipate the future movements of other traffic participants, and make robust sequential decisions to avoid collisions while progressing toward a destination. Here, we process 27K and 654K samples sourced from nuScenes (Caesar et al., 2020a) and OpenScene (Contributors, 2023), respectively. We directly record the original multi-view images, instructions, and vehicle state information from the dataset. Note that we do not collect explicit surrounding information (such as lane details), in contrast to common autonomous driving baselines (Chen et al., 2024c; Li et al., 2024b).

**Web-Video Navigation.** (2.03M) We also leverage the Sekai dataset (Li et al., 2025a), which provides a collection of approximately 182K YouTube videos with corresponding instructions (generated by VLMs (Bai et al., 2025)) and trajectories (generated by SLAM systems (Li et al., 2025b)). Although these navigation samples contain imperfect instructions and trajectories, they remain valuable for incorporating real-world navigation scenarios. Similar findings have been reported in (Cheng et al., 2025; Wei et al., 2025).

**Open-World Question-Answering.** (4.76M) Following existing video-based VLMs (Li et al., 2023; Shen et al., 2024; Wang et al., 2025c), we collect 3.15M image QA samples and 1.61M video QA samples, which encompass rich and comprehensive knowledge for open-world understanding.

## A.5 DISCRETE ACTION PROCESSING

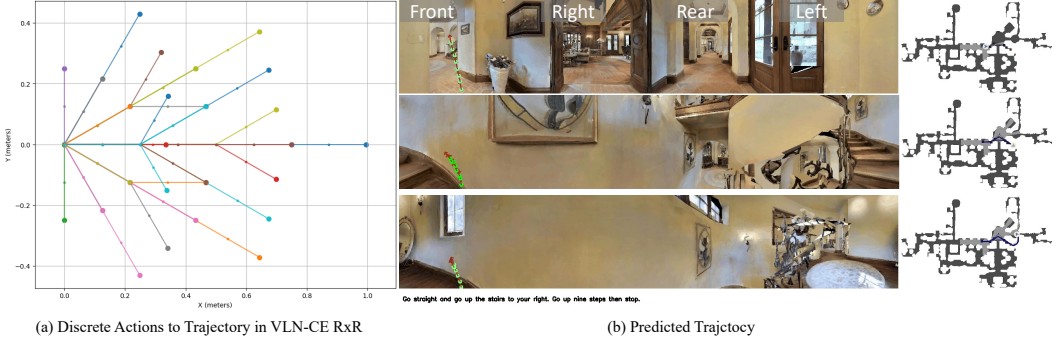

(a) Discrete Actions to Trajectory in VLN-CE RxR        (b) Predicted Trajctocy

Figure 9: Visualization of the trajectory (VLN-CE RxR) for (a) training and (b) evaluation.

For navigation tasks built on the Habitat environment (Savva et al., 2019a), which utilizes low-level discrete actions such as `Move_Forward`, `Turn_Left`, `Turn_Right`, and `Stop`. However, the definitions of these discrete actions vary slightly across different navigation tasks. For example, in VLN-CE R2R, `Turn_Left` indicates a 15-degree turn, whereas in VLN-CE RxR and HM3D-ObjNav, it indicates a 30-degree turn. To unify all navigation tasks with discrete actions, we employ a simple strategy: we consider moving forward by 12.5 cm or turning by 15 degrees as an atomic

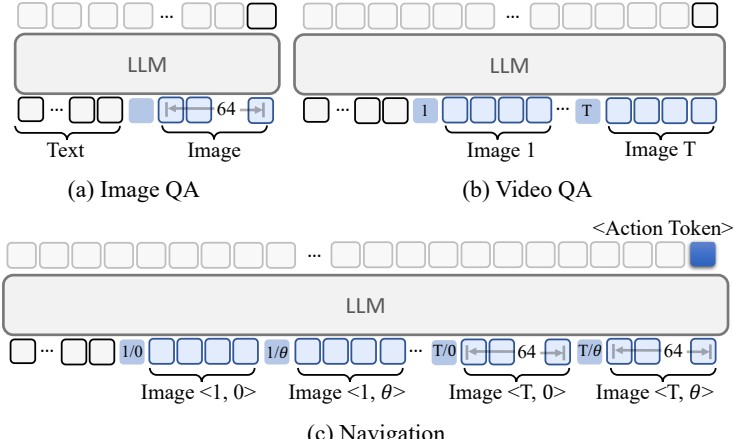

(a) Image QA

(b) Video QA

(c) Navigation

Figure 10: **Token Organization Strategy of NavFoM Across Different Tasks.** (a) For image question answering, fine-grained visual tokens are utilized, incorporating only the base embedding of TVI tokens. (b) For video question answering, coarse-grained visual tokens are employed, which include both the base embedding and the time embedding of TVI tokens. (c) For navigation, both coarse-grained and fine-grained visual tokens are used, integrating the base, time, and angle embeddings of TVI tokens.

operation. We then construct the trajectory based on the accumulation of these atomic operations. Although the resulting trajectory could be zigzag (Figure 9), after fine-tuning on all navigation datasets, we find that the predicted trajectory of our method is smooth and meaningfully directed toward the target.

### A.6 TOKEN ORGANIZATION

We provide a detailed illustration of the token organization strategy for different tasks in Figure 10. For Image QA, we use $E_{\text{Base}}$ along with fine-grained visual tokens (64 tokens per image) to represent the image. For Video QA, we incorporate $\mathbf{E}_{\text{Base}} + \mathcal{P}_{\text{time}}(\text{TimePE}(t))$ to encode temporal information for each frame, and employ coarse-grained visual tokens (4 tokens per frame) to avoid an excessive number of tokens.

## B EXPERIMENT DETIALS

### B.1 BENCHMARKS

We give a detailed introduction to evaluation benchmarks:

- **Vision-and-Language Navigation**: We evaluate our method on the VAL-Unseen splits of the VLN-CE R2R (Krantz et al., 2020) and RxR (Ku et al., 2020a) benchmarks, which require the robot to follow instructions in unseen indoor environments. We also evaluate our method on the Open-UAV benchmark (Wang et al., 2024a), which requires the UAV to follow instructions in unseen outdoor environments.

- **Object goal navigation**: We follow previous methods (Zhang et al., 2025a; Zhu et al., 2025) to evaluate the generalizability of object-goal navigation on the HM3D-OVON dataset (Yokoyama et al., 2024b), an open-vocabulary object navigation benchmark, in a zero-shot manner.

- **Active Visual Tracking**: We evaluate our method on EVT-Bench (Wang et al., 2025c), a challenging benchamrak requrie the robot to distinguish and follow target within crowded environments.

- **Autonomous Driving**: We evaluate our method on mainstream benchmarks, namely nuScenes (Caesar et al., 2020a) and NAVSIM (Dauner et al., 2024b), for open-loop and pseudo-simulation evaluation. Our evaluation strategy is consistent with existing baseline (Liao et al., 2024a) to ensure a fair comparison.

### B.2 METRICS

**Success Related Metrics.** We report three success-related metrics (Anderson et al., 2018): Navigation Error (NE) measures the average distance between the agent's final position and the goal; Success Rate (SR) calculates the percentage of episodes where the agent stops within a threshold distance of the goal, while additionally requiring the goal to be within the agent's receptive field for OVON and EVT-Bench; and Oracle Success (OS) reports the percentage of episodes where the agent passes within the threshold distance at any timestep. Success thresholds vary across benchmarks, and we follow their default settings: 0–3m for VLN-CE R2R, RxR, HM3D-OVON; 1–3m for EVT-Bench; and 0–20m for Open-UAV.

**Trajectory Quality Metrics.** To account for path efficiency, we measure Success weighted by Path Length (SPL) (Batra et al., 2020), which rewards successful agents that adhere closer to the optimal path length: $\mathrm{SPL} = \frac{1}{N} \sum_{i=1}^{N} S_i \cdot \frac{L_i^\star}{\max(L_i, L_i^\star)}$, where $S_i$ indicates success for episode $i$, $L_i^\star$ is the shortest-path distance, and $L_i$ is the executed path length; We also report normalized Dynamic Time Warping (nDTW) (Ilharco et al., 2019), which quantifies the fidelity of the agent's path relative to the ground truth trajectory: $\mathrm{nDTW} = \exp\left(-\frac{\mathrm{DTW}(\tau, \hat{\tau})}{\eta}\right)$, where $\mathrm{DTW}(\tau, \hat{\tau})$ is the dynamic time warping distance (Berndt & Clifford, 1994) between reference path $\tau$ and predicted path $\hat{\tau}$, and $\eta$ is the shortest-path distance from start to goal. Specifically for the tracking task, we use the Tracking Rate (TR) (Puig et al., 2023), which measures the agent's temporal consistency, defined as the proportion of steps where the target is maintained within the sensor's field of view and a 1–3m range relative to the total episode length.

**Autonomous Driving Evaluations.** For the autonomous driving evaluation, we report L2 distance and Collision Rate (CR) for open-loop planning (Caesar et al., 2020a). L2 measures the average Euclidean distance between the predicted and ground truth waypoints, while CR measures the frequency of intersection with obstacles. For closed-loop evaluation in NAVSIM, we use the PDM score (PDMS) (Dauner et al., 2024b). PDMS is a holistic metric composed of weighted sub-scores: No at-fault Collisions (NC) and Drivable Area Compliance (DAC) penalize critical safety infractions; Time-to-Collision (TTC) and Comfort (Comf.) assess interaction safety and ride smoothness; and Ego Progress (EP) measures the distance traveled along the route as a ratio to a safe upper bound.

### B.3 TRAINING STRATEGY

**Accelerating Training by Caching Visual Features.** Due to the long horizon of videos (hundreds of frames), encoding all images online in a large batch can be computationally expensive. To mitigate this issue, we leverage a visual feature caching mechanism (Yan et al., 2022) and construct a visual feature database (See Figure 11). Note that we only cache coarse-grained visual tokens (4 tokens per frame), which require significantly less disk space compared to storing full videos, as a single episode of navigation typically produces dozens of videos. For image QA and the latest observation in navigation, we still use visual encoders online to extract fine-grained visual tokens (64 tokens per frame). This approach reduces training time (2.9× faster) and GPU memory usage (1.8× less).

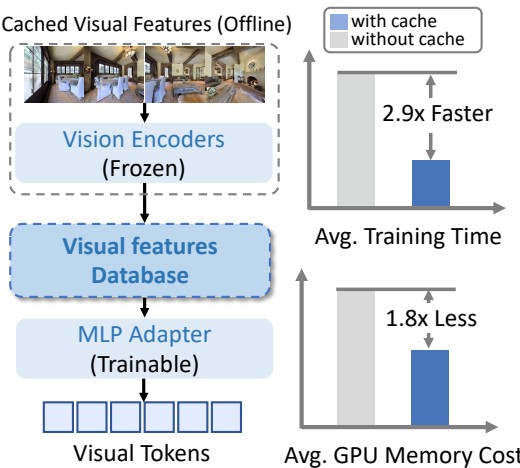

Figure 11: **Offline Visual Feature Cached.** We precomputed video frames and navigation hisitroy and saved as corase visual tokens.

### B.4 REAL-WORLD DEPLOYMENT SYSTEM

We regard our model as a general Visual-Language-Action (VLA) model capable of

Table 6: **Comprehensive results on `OpenUAV` benchmark with L1 level assistant. `Seen`** denotes the seen split, while **`UO`** and **`UM`** represent the Test Unseen Object Set and Test Unseen Map Set respectively. DA refers to a model trained using backtracking sampling-based data aggregation. The **best** and the second best results are denoted by **bold** and underline.

| Method | Test Set | Full | | | | Easy | | | | Hard | | | |
|---|---|---|---|---|---|---|---|---|---|---|---|---|---|
| | | NE↓ | SR↑ | OSR↑ | SPL↑ | NE↓ | SR↑ | OSR↑ | SPL↑ | NE↓ | SR↑ | OSR↑ | SPL↑ |
| *OpenUAV Seen Set* | | | | | | | | | | | | | |
| Human | Seen | 14.15 | 94.51 | 94.51 | 77.84 | 11.68 | 95.44 | 95.44 | 76.19 | 17.16 | 93.37 | 93.37 | 79.85 |
| Random Action | Seen | 222.20 | 0.14 | 0.21 | 0.07 | 142.07 | 0.26 | 0.39 | 0.13 | 320.12 | 0.00 | 0.00 | 0.00 |
| Fixed Action | Seen | 188.61 | 2.27 | 8.16 | 1.40 | 121.36 | 3.48 | 11.48 | 2.14 | 270.69 | 0.79 | 4.09 | 0.49 |
| CMA (Wang et al., 2024a) | Seen | 135.73 | 8.37 | 18.72 | 7.90 | 84.89 | 11.48 | 24.52 | 10.68 | 197.77 | 4.57 | 11.65 | 4.51 |
| TravelUAV (Wang et al., 2024a) | Seen | 106.28 | 16.10 | 44.26 | 14.30 | 68.78 | 18.84 | 47.61 | 16.39 | 152.04 | 12.76 | 40.16 | 11.76 |
| TravelUAV-DA | Seen | 98.66 | 17.45 | 48.87 | 15.76 | 66.40 | 20.26 | 51.23 | 18.10 | 138.04 | 14.02 | **45.98** | 12.90 |
| **NavFoM (Four views)** | Seen | **93.05** | **29.17** | **49.24** | **25.03** | **58.98** | **32.91** | **53.16** | **27.87** | 143.83 | **23.58** | 43.40 | **20.80** |
| *OpenUAV Unseen Set* | | | | | | | | | | | | | |
| Random Action | UO | 260.14 | 0.16 | 0.16 | 0.16 | 174.10 | 0.48 | 0.48 | 0.48 | 302.96 | 0.00 | 0.00 | 0.00 |
| Fixed Action | UO | 212.84 | 3.66 | 9.54 | 2.16 | 151.66 | 6.70 | 13.88 | 3.72 | 243.29 | 2.14 | 7.38 | 1.38 |
| CMA (Wang et al., 2024a) | UO | 155.79 | 9.06 | 16.06 | 8.68 | 102.92 | 14.83 | 22.49 | 13.90 | 182.09 | 6.19 | 12.86 | 6.08 |
| TravelUAV (Wang et al., 2024a) | UO | 118.04 | 22.42 | 46.90 | 20.51 | 86.12 | 24.40 | 49.28 | 22.03 | 134.03 | 21.43 | 45.71 | 19.75 |
| **NavFoM (Four views)** | UO | **108.04** | **29.83** | **47.99** | **27.20** | **70.51** | **32.54** | **50.72** | **29.54** | **133.01** | **28.03** | **46.18** | **25.64** |
| Random Action | UM | 202.98 | 0.00 | 0.00 | 0.00 | 158.46 | 0.00 | 0.00 | 0.00 | 265.88 | 0.00 | 0.00 | 0.00 |
| Fixed Action | UM | 180.47 | 0.52 | 2.61 | 0.39 | 132.89 | 0.89 | 4.28 | 0.67 | 247.72 | 0.00 | 0.25 | 0.00 |
| CMA (Wang et al., 2024a) | UM | 141.68 | 2.30 | 10.02 | 2.16 | **102.29** | 3.57 | 14.26 | 3.33 | 197.35 | 0.50 | 4.03 | 0.50 |
| TravelUAV (Wang et al., 2024a) | UM | 138.80 | 4.18 | **20.77** | 3.84 | 102.94 | 4.63 | **22.82** | 4.24 | 189.46 | 3.53 | **17.88** | 3.28 |
| **NavFoM (Four views)** | UM | **125.10** | **6.30** | 18.95 | **5.68** | 102.41 | **6.77** | 20.07 | **6.04** | **170.58** | **5.36** | 15.71 | **4.97** |

driving different embodiments to complete various navigation tasks. To achieve this, our model takes visual observations—obtained from one or more cameras—along with instructions, and directly predicts a trajectory. We then utilize off-the-shelf APIs (which may include Lidar or other sensors if necessary) specific to each embodiment to drive the robot along the predicted trajectory.

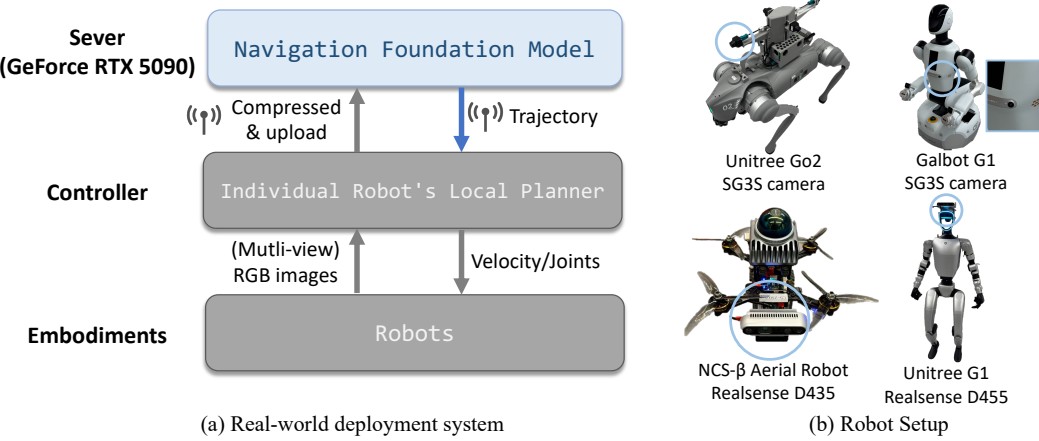

(a) Real-world deployment system          (b) Robot Setup

Figure 12: **Real-world deployment setup.** We provide the system architecture of our methods and the corresponding robots that were tested in the paper.

An illustration of our real-world system is provided in Figure 12. Specifically, we deploy our model on a remote server equipped with a GeForce RTX 5090 GPU and use the Internet for communication between the server and the client (which includes the controller and embodiments). Given a user instruction, the robots compress their current observations and transmit them to the server. The server then processes both the observations and the instruction to output a trajectory. This trajectory is subsequently processed by the local planner of each individual robot, which sends appropriate commands (*e.g.*, velocity or joint controls) to drive the robot.

Table 7: **Object goal navigation.** Comparison on HM3D-OVON Yokoyama et al. (2024b). *: denotes zero-shot evaluation. We report the performance of our method on egocentric and four-view settings. The **best** and the second best results are denoted by **bold** and underline.

| Method | VAL SEEN | | VAL SEEN SYNONYMS | | VAL UNSEEN | |
| --- | --- | --- | --- | --- | --- | --- |
| | SR↑ | SPL↑ | SR↑ | SPL↑ | SR↑ | SPL↑ |
| BC | 11.1 | 4.5 | 9.9 | 3.8 | 5.4 | 1.9 |
| DAgger | 11.1 | 4.5 | 9.9 | 3.8 | 5.4 | 1.9 |
| RL | 18.1 | 9.4 | 15.0 | 7.4 | 10.2 | 4.7 |
| BCRL | 39.2 | 18.7 | 27.8 | 11.7 | 18.6 | 7.5 |
| DAgRL | 41.3 | 21.2 | 29.4 | 14.4 | 18.3 | 7.9 |
| VLFM* (Yokoyama et al., 2024a) | 35.2 | 18.6 | 32.4 | 17.3 | 35.2 | 19.6 |
| DAgRL+OD (Yokoyama et al., 2024b) | 38.5 | 21.1 | 39.0 | 21.4 | 37.1 | 19.8 |
| Uni-NaVid* (Zhang et al., 2025a) | 41.3 | 21.1 | 43.9 | 21.8 | 39.5 | 19.8 |
| MTU3D (Zhu et al., 2025) | **55.0** | 23.6 | 45.0 | 14.7 | 40.8 | 12.1 |
| **NavFoM * (Single view)** | 37.7 | 25.5 | 43.3 | 29.9 | 43.6 | 31.3 |
| **NavFoM * (Four views)** | 40.1 | **27.1** | **45.4** | **32.6** | **45.2** | **31.9** |

# C ADDITIONAL EXPERIMENTS

## C.1 PERFORMANCE ON OPENUAV

We report the performance of our method in a challenging UAV scenario (Wang et al., 2024a) in Table 6, which requires the UAV to follow natural language instructions and execute long-horizon trajectories (averaging 200 meters) to reach described targets in outdoor environments. Note that our method uses trajectories directly collected from the TravelUAV (Wang et al., 2024a) training split (mimicking ground truth trajectories), as no strong baseline was available to collect expert trajectories as was done for the ObjectNav data collection. Despite this, our approach achieves state-of-the-art performance compared to prior UAV-specific baselines such as TravelUAV, without relying on downward-facing cameras as used in those methods (we plan to incorporate additional degrees of freedom in camera configurations in future work). This clearly demonstrates the effectiveness of our approach and the benefits of learning from diverse navigation tasks (Figure 7).

However, we observe that all methods perform poorly on the Unseen-Map split, which requires an average traversal of 300 meters through complex neighborhoods to reach unseen targets. This is because the unseen split demands more advanced navigation capabilities, such as efficient exploration of large-scale environments, which in turn relies on higher-quality UAV data.

## C.2 PERFORMANCE ON OVON

Following prior work (Zhang et al., 2025a; Zhu et al., 2025), we evaluate search capability on an open-vocabulary benchmark (Yokoyama et al., 2024b) under a zero-shot setting. The results are presented in Table 7, which includes performance for both single-camera and four-camera configurations. Under the single-camera setting, our method achieves performance comparable to that of the state-of-the-art (SOTA) approach (Zhu et al., 2025) on both the VAL SEEN and VAL SEEN SYNONYMS splits in a zero-shot evaluation setting. On the more challenging VAL UNSEEN split, our method outperforms the SOTA method, improving the success rate (SR) from $40.8\%$ to $43.6\%$. Furthermore, when transitioning from the single-camera to the four-camera setting, we observe consistent improvements across all splits and metrics. Notably, our model was trained only on single-camera search samples, demonstrating that co-tuning across different camera configurations enhances generalization to varied camera setups.

## C.3 PERFORMANCE ON THE EVT-BENCH

We evaluate our method on EVT-Bench (Wang et al., 2025c) (including both the Single Target and Distracted Target splits) under both single-view and four-view camera settings (Table 8). Note that our model is trained only on the single-view setting and evaluated on the four-view setting in a zero-shot manner. Our results demonstrate that the proposed method achieves state-of-the-art (SOTA) performance under the single-view setting, outperforming the previous baseline, TrackVLA (Wang et al., 2025c), which was specifically fine-tuned on tracking data. Furthermore, when the camera

Table 8: **Performance on EVT-Bench**. †: Uses GroundingDINO (Liu et al., 2023b) as the open-vocabulary detector. ‡: Uses SoM (Yang et al., 2023)+GPT-4o (OpenAI, 2024) as the visual foundation model. The **best** and the second best results are denoted by **bold** and underline.

| Method | Single Target | | Distracted Target | |
|---|---|---|---|---|
| | SR↑ | TR↑ | SR↑ | TR↑ |
| IBVS† (Gupta et al., 2016) | 42.9 | 56.2 | 10.6 | 28.4 |
| PoliFormer† (Zeng et al.) | 4.67 | 15.5 | 2.62 | 13.2 |
| EVT (Zhong et al., 2024) | 24.4 | 39.1 | 3.23 | 11.2 |
| EVT‡ (Zhong et al., 2024) | 32.5 | 49.9 | 15.7 | 35.7 |
| Uni-NaVid (Zhang et al., 2025a) | 25.7 | 39.5 | 11.3 | 27.4 |
| TrackVLA (Wang et al., 2025c) | 85.1 | 78.6 | 57.6 | 63.2 |
| **NavFoM (Single view)** | 85.0 | 80.5 | 61.4 | **68.2** |
| **NavFoM (Four views)** | **88.4** | **80.7** | **62.0** | 67.9 |

Table 9: **Comparison on planning-oriented NAVSIM `navtest` split with closed-loop metrics.** $\mathcal{V}_{8192}$ denotes 8192 anchors. The **best** and the second best results are denoted by **bold** and underline.

| Method | Observation & Structure | | | Metrics | | | | | |
|---|---|---|---|---|---|---|---|---|---|
| | Camera | Lidar | VLM-Based | NC ↑ | DAC ↑ | TTC ↑ | Comf. ↑ | EP ↑ | PDMS ↑ |
| Human | - | - | - | 100 | 100 | 100 | 99.9 | 87.5 | 94.8 |
| Constant Velocity | - | - | - | 69.9 | 58.8 | 49.3 | 100 | 49.3 | 21.6 |
| Ego Status MLP | - | - | - | 93.0 | 77.3 | 83.6 | 100 | 62.8 | 65.6 |
| LTF (Chitta et al., 2022) | ✓ | ✓ | - | 97.4 | 92.8 | 92.4 | **100** | 79.0 | 83.8 |
| Transfuser (Chitta et al., 2022) | ✓ | ✓ | - | 97.7 | 92.8 | 92.8 | **100** | 79.2 | 84.0 |
| VADv2-$\mathcal{V}_{8192}$ (Chen et al., 2024b) | ✓ | ✓ | - | 97.2 | 89.1 | 91.6 | **100** | 76.0 | 80.9 |
| Hydra-MDP-$\mathcal{V}_{8192}$ (Li et al., 2024c) | ✓ | ✓ | - | 97.9 | 91.7 | 92.9 | **100** | 77.6 | 83.0 |
| DiffusionDrive (Liao et al., 2024a) | ✓ | ✓ | - | **98.2** | **96.2** | 94.7 | **100** | **82.2** | **88.1** |
| DRAMA (Yuan et al., 2024) | ✓ | ✓ | ✓ | 98.0 | 93.1 | **94.8** | **100** | 80.1 | 85.5 |
| UniAD (Hu et al., 2023) | ✓ | - | - | 97.8 | 91.9 | 92.9 | **100** | 78.8 | 83.4 |
| PARA-Drive (Weng et al., 2024) | ✓ | - | - | 97.9 | 92.4 | 93.0 | 99.8 | 79.3 | 84.0 |
| LAW (Li et al., 2024b) | ✓ | - | - | 96.4 | **95.4** | 88.7 | 99.9 | **81.7** | **84.6** |
| DrivingGPT (Chen et al., 2024c) | ✓ | - | ✓ | **98.9** | 90.7 | **94.9** | 95.6 | 79.7 | 82.4 |
| **NavFoM (Eight views)** | ✓ | - | ✓ | 97.7 | 93.5 | 92.3 | **100** | 79.6 | 84.3 |

setup is increased from single-view to four-view (in a zero-shot manner), our method continues to improve its performance. However, compared to the improvement observed in VLN (a 6.8% ↑ in SR on VLN-CE RxR), the gains here are relatively modest (0.6% ↑ in SR). We attribute this to the fact that most targets in EVT-Bench are spawned in front of the robot, a key assumption of this benchmark. We plan to further investigate this issue through both simulation and methodological enhancements, such as incorporating randomly positioned surrounding targets in future work.

## C.4 PERFOMRENCE ON NAVSIM

We conduct experiments to evaluate our method on eight-view settings autonoums driving (without fine-tuning for specific configurations). Results on NAVSIM in Table 9. We observe that our method achieves performance comparable to SOTA methods on both benchmarks, without explicitly modeling driving-related information such as lane markings, nearby vehicles, or other contextual elements. We believe our approach can be further improved by incorporating scene descriptions as prompts, similar to other baseline methods. We are also interested in evaluating this model in closed-loop autonomous driving simulators such as (Dosovitskiy et al., 2017).

Table 11: **Computational cost.** We report the converged mem cost and converged inference speed.

| Model Version | Converged Mem Cost | Converged Inference |
|---|---|---|
| RTX 4090 original (16bit) | 19.8 GB | 218 ms |
| RTX 4090 Quantized (4bit) | 10.7 GB | 248 ms |
| Jetson Thor Quantized (16bit) | 19.1 GB | 566 ms |

## C.5 PERFOMRENCE ON NUSCENE

Table 10: **Comparison on planning-oriented `nuScenes` dataset with open-loop metrics.** Metric calculation follows DiffusionDrive (Liao et al., 2024b). The **best** and the second best results are denoted by **bold** and underline.

| Method | Observation & Structure | | L2 (m) ↓ | | | | Collision (%) ↓ | | | |
|---|---|---|---|---|---|---|---|---|---|---|
| | Camera | VLM-Based | 1s | 2s | 3s | Avg. | 1s | 2s | 3s | Avg. |
| ST-P3 (Hu et al., 2022) | ✓ | - | 1.33 | 2.11 | 2.90 | 2.11 | 0.23 | 0.62 | 1.27 | 0.71 |
| UniAD (Hu et al., 2023) | ✓ | - | 0.45 | 0.70 | 1.04 | 0.73 | 0.62 | 0.58 | 0.63 | 0.61 |
| VAD (Jiang et al., 2023) | ✓ | - | 0.41 | 0.70 | 1.05 | 0.72 | 0.07 | 0.17 | 0.41 | 0.22 |
| SparseDrive (Sun et al., 2024) | ✓ | - | 0.29 | 0.58 | 0.96 | 0.61 | 0.01 | **0.05** | 0.18 | **0.08** |
| DiffusionDrive (Liao et al., 2024b) | ✓ | - | 0.27 | 0.54 | 0.90 | 0.57 | 0.03 | **0.05** | **0.16** | **0.08** |
| DriveVLM (Tian et al., 2024) | ✓ | ✓ | 0.18 | 0.34 | 0.68 | 0.40 | 0.10 | 0.22 | 0.45 | 0.27 |
| EMMA (Hwang et al., 2024) | ✓ | ✓ | **0.14** | **0.29** | **0.54** | **0.32** | - | - | - | - |
| DME-Driver (Han et al., 2025) | ✓ | ✓ | 0.45 | 0.91 | 1.58 | 0.98 | 0.05 | 0.28 | 0.55 | 0.29 |
| Omni-Q (Wang et al., 2025d) | ✓ | ✓ | **0.14** | **0.29** | 0.55 | 0.33 | **0.00** | 0.13 | 0.78 | 0.30 |
| Omni-L (Wang et al., 2025d) | ✓ | ✓ | 0.15 | 0.36 | 0.70 | 0.40 | 0.06 | 0.27 | 0.72 | 0.35 |
| ORION (Fu et al., 2025a) | ✓ | ✓ | 0.17 | 0.31 | 0.55 | 0.34 | 0.05 | 0.25 | 0.80 | 0.37 |
| **NavFoM (Six views)** | ✓ | ✓ | 0.26 | 0.39 | 0.60 | 0.42 | 0.07 | 0.11 | 0.18 | 0.12 |

We report the performance of our method on six-camera setting autonomous driving benchmark nuScene (Caesar et al., 2020b) in Table 10. We compare our method with strong baselines that are specifically designed for autonomous driving. Nevertheless, our method achieves comparable performance to these methods without explicitly modeling driving-related information.

## C.6 VISUAL RESULTS OF SYNTHETIC ENVIRONMENTS

We provide visual results on benchmarks in Figure 13 from VLN-CE RxR (Ku et al., 2020a), EVT-Bench (Wang et al., 2025c), OVON (Yokoyama et al., 2024b), openUAV (Wang et al., 2024a), nuScenes (Caesar et al., 2020a) and NAVSIM (Dauner et al., 2024a).

# D ABLATION STUDY

**Performance on differnt number of cameras.** We evaluate the effectiveness of incorporating additional cameras in navigation tasks on VLN-CE RxR, a benchmark that offers a relatively comprehensive suite of vision-language navigation challenges. The results are presented in Table 14, which compares configurations of one, two, three, four, and six cameras mounted around the robot to achieve a wider field of view. We observe consistent performance improvements when increasing the number of cameras from one to four, validating that enhanced environmental observations contribute positively to navigation performance. Notably, however, expanding to six cameras leads to a slight degradation in performance. We attribute this to the fact that six cameras do not provide substantially more observational coverage compared to four cameras, while the increased number of view tokens reduces the capacity available for encoding historical frames (Equation 5). This weaks the alignment between the navigation history and the instruction. We suggest that this issue could be mitigated by adopting an adaptive multi-view token encoding strategy. To maintain coherence in the current work, we leave this exploration for future research.

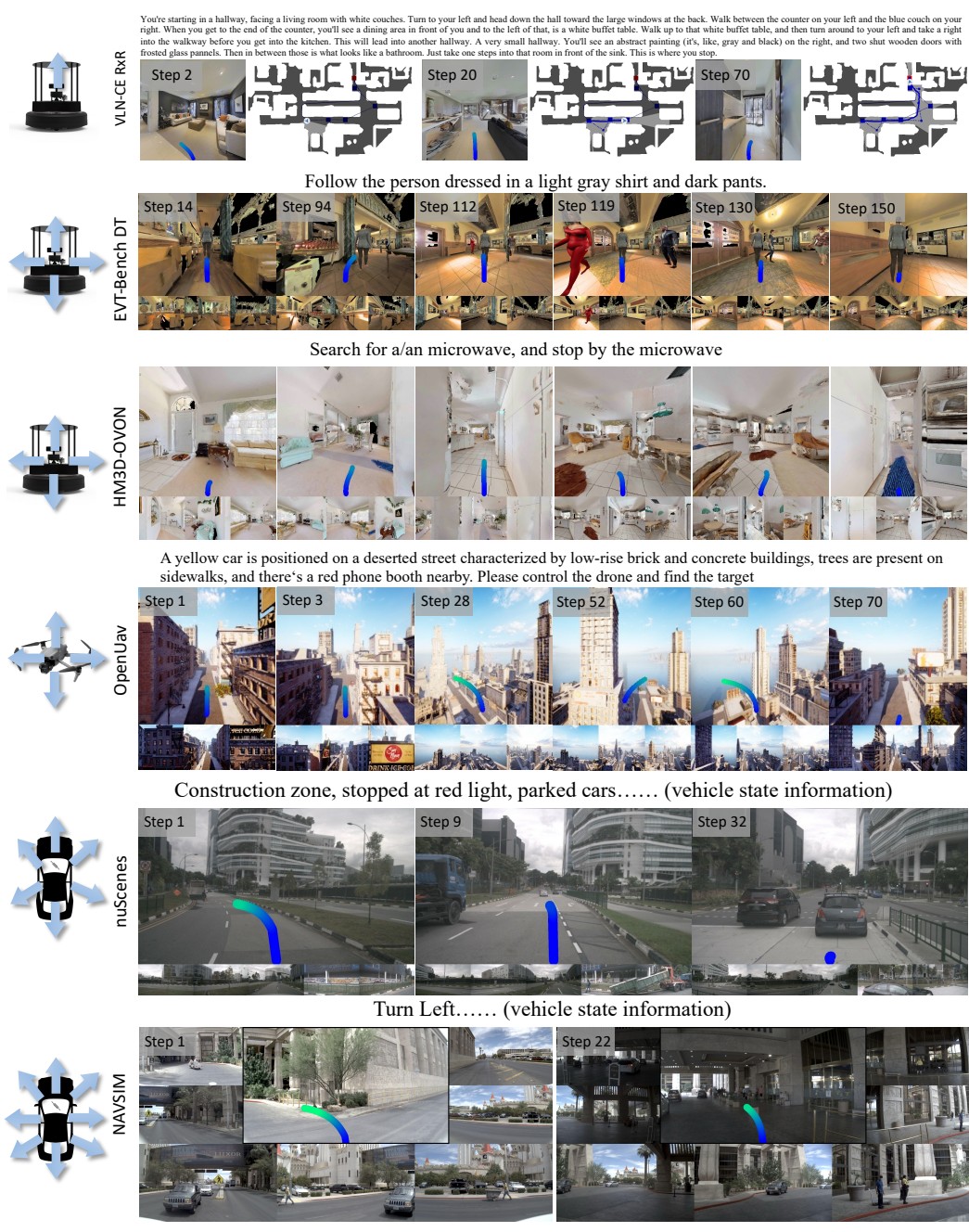

Figure 13: **Visualization of perfomrence on benchmarks.** We report visual results of NavFoM on VLN-CE RxR (single-view), EVT-Bench Distracted Targets (four-view), OpenUAV (four-view), NeuScenes (six-view), OpenScenses (Eight-view).

# E   REAL-WORLD EXPERIMENTS

**Real-world deployment cost.** We have conducted additional experiments on deployment costs. Specifically, we provide the original costs (16-bit) and quantized version (4-bit) of our model ('7B LLM + 2B ViT, 2048 Token Budget, four-camera view') on VLN-CE RxR in the table. The results can be found in Table 11. We find that our quantized models (4-bit via Bitsandbytes[*]) significantly

---
[*] https://huggingface.co/docs/transformers/quantization/bitsandbytes

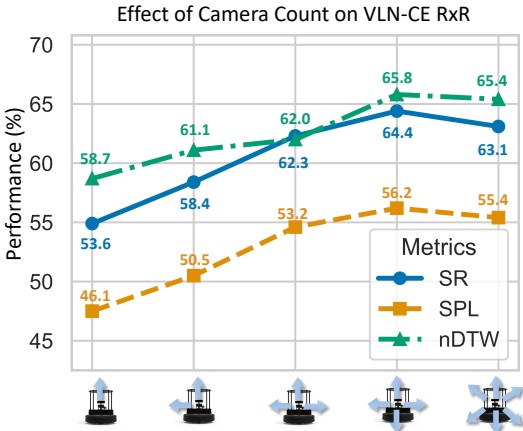

Figure 14: **Ablation study on the number of cameras in VLN-CE RxR.** We report the performance under five different camera configurations (from left to right: one-, two-, three-, four-, and six-camera settings), with same token budget ($B = 2048$).

reduce deployment costs (using only 53.9% of the original memory) while maintaining comparable performance. This also enables our method to be deployed on the latest onboard GPUs. As an example, we deployed the 16-bit quantized model on a Jetson Thor and observed stable performance, with an average inference speed of 566 ms per trajectory prediction.

Regarding deployment memory cost, techniques such as SmoothQuant (Xiao et al., 2023a) and quantization-aware training (Dettmers et al., 2023) could significantly reduce memory usage while maintaining strong performance. For inference speed, there are also existing advanced techniques such as LLM streaming (Xiao et al., 2023b) (which is suitable for processing online captured video in robot tasks) and the Speculative Decoding strategy Leviathan et al. (2023). These methods have demonstrated significant inference speed improvements in complicated tasks (Leviathan et al., 2023). In summary, we believe that with the rapid development of graphics hardware and acceleration methods, fast and convenient deployment of large model-based approaches will become a promising direction.

**Real-world performance on 110 reproducible test cases.** To evaluate the real-world performance of our method, we designed a series of navigation test cases with different capabilities (including 50 VLN samples, 30 search samples, and 30 tracking samples). Specifically, we constructed a 5m × 5m space and recorded the locations of the robot, obstacles, and targets for each test case. We report both qualitative and quantitative results of NavFoM in complex scenarios across these navigation capabilities. The results are presented in Figure 15. Our findings indicate that NavFoM demonstrates strong real-world performance: it correctly understands the surrounding environment and plans appropriate trajectories to accomplish the task. Moreover, compared to the strong baseline Uni-NaVid (Zhang et al., 2025a), our method exhibits significant improvements across both tasks, demonstrating its superior performance in real-world environments.

**Visual results of challenging cross-task and cross-emdbodiement real-world experiments.** We also conduct extensive experiments on more challenging scenarios with different embodiments (quadruped robots, humanoids, drones, and wheeled robots). The results are shown in Figure 16, where we find that our method can handle complicated real-world environments and fulfill long-horizon instructions. We encourage readers to view our accompanying videos for a more intuitive demonstration.

## F  FAILURE CASE ANALYSIS

We provide a more detailed analysis of the failure cases, covering both benchmark and real-world environments.

**Benchmark Environments**: We analyze benchmark failure cases in in VLN-CE RxR, the limited field of view (FoV) in the single-camera setup significantly affects the ability to ground visual in-

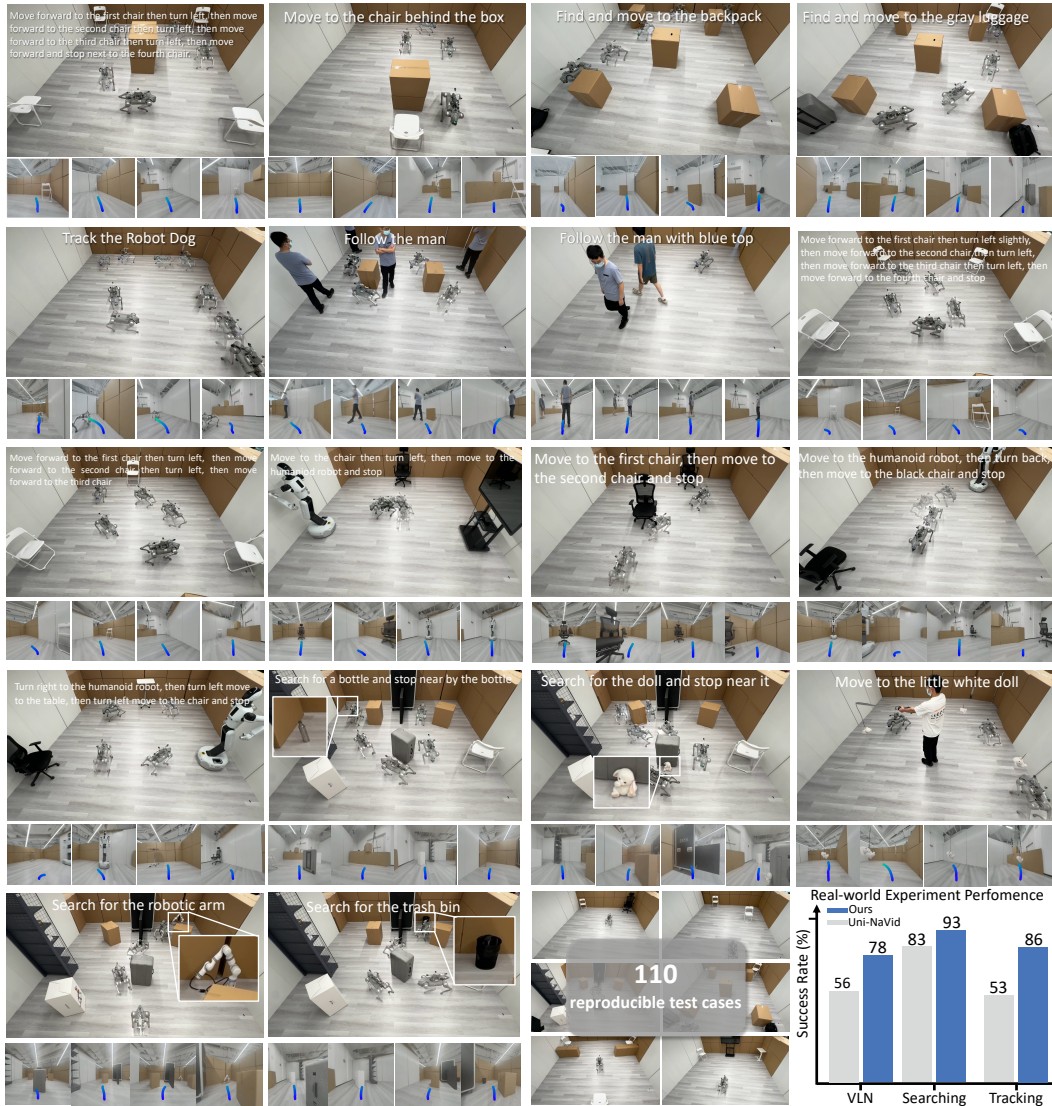

Figure 15: **Real-world experiments.** We report both the qualitive and quantitive results of NavFoM on complex seniors among different navigation cabability.

formation with instructions. When switching to a four-camera setting (360° FoV), the success rate increases from 57.4% to 64.4%. We observe that about 51% of failures are due to dataset/simulator problems, including rendering quality and misleading instructions (e.g., ambiguous landmarks). The remaining failures stem from model capability issues (49%), such as failing to align history with instructions (e.g., performing early stops) or failing to execute sufficient turns (especially at challenging narrow corners). This indicates that future efforts should focus on improving both datasets/simulators and model capabilities.

**Real-world Environments**: During the real-world experiments, we find that most failure cases stem from recognizing small objects (such as bottles or books) from a long distance or understanding blurred images while the robot is moving. Additionally, extremely challenging scenarios, such as following long-horizon instructions (thousands of words) or searching for an object within a very large building (hundreds of square meters), pose critical challenges to the method. We believe that a more robust real-world approach requires collaborative efforts in both model capabilities (perception, reasoning, memory) and hardware components (camera, computational resources).

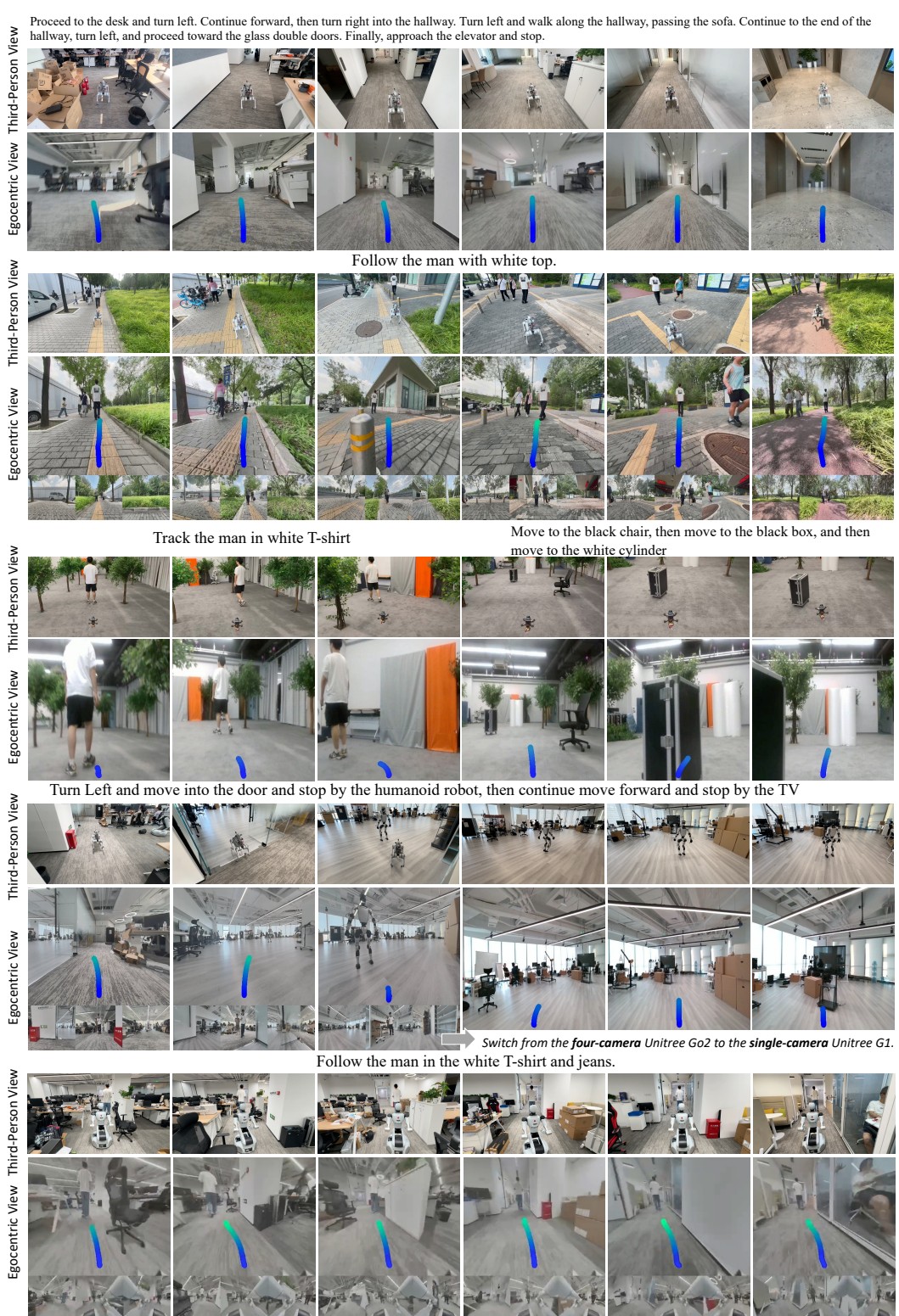

Figure 16: Visualization of real-world experiments on cross-task and cross-embodiment settings.

