# OpenReview forum: "Embodied Navigation Foundation Model"
_ICLR.cc/2026/Conference — ICLR 2026 Poster_

### Official Review · Reviewer_iLxn · 2025-10-26

**Soundness:** 3
**Presentation:** 3
**Contribution:** 3
**Rating:** 6
**Confidence:** 3

**Summary:**

The paper proposes a cross-task, cross-embodiment navigation foundation model that ingests egocentric multi-view video plus language and outputs waypoint trajectories, trained on 8M navigation samples spanning quadrupeds, drones, wheeled robots, and vehicles.

Its key innovations are Temporal-Viewpoint Indicator tokens to encode camera view and time, and a Budget-Aware Temporal Sampling scheme that keeps recent context while respecting a token budget.

**Strengths:**

* The paper introduce Temporal-Viewpoint Indicator, which provide explicit view and time encoding so the LLM can align frames across cameras, and Budget-Aware Temporal Sampling preserves recent context under a fixed token budget.
* Across seven public benchmarks and additional real-world robot trials, NavFoM attains state-of-the-art or highly competitive results without task-specific fine-tuning.

**Weaknesses:**

* The model fine-tunes only a single epoch, this suggests possible under-optimization given the 12.7M sample corpus.
* Multi-task training uses a fixed loss weight 𝜙 = 10 to scale navigation MSE against QA loss, the paper doesn’t probe sensitivity to this choice.

**Questions:**

An ablation study on the loss weights can help validate the effectiveness of the fixed hyperparameters.

---

> ### Author Response · Authors · 2025-11-21
> **Rebuttal**
>
> We are delighted that the reviewer iLxn finds our method to be innovative, technically sound, and effective with strong results on both benchmarks and real-world applications. We also thank the reviewer for the constructive feedback, which has helped us revise and strengthen our paper. We hope the following clarifications address the reviewer's remaining concerns:
>
> ### 1. **One epoch of training**
>
> Training for one epoch is a common practice in large model-based methods [1,2]. Since training large models often requires a substantial amount of data (12M used in our method), training for more epochs can lead to significantly higher costs. Here, we also conduct an ablation study comparing one epoch and two epochs on VLN-CE RxR. Due to the high training cost, we perform experiments using half of the data (6M), which still constitutes a large dataset. The results show that adding one more epoch does not yield significant improvements. Furthermore, we find that `increasing the scale of data is more beneficial than training for an additional epoch`. If the reviewer considers it necessary, we would be happy to conduct a full-data training experiment and include the results in the camera-ready version accordingly.
>
>
> | Model Version  |  Success Rate (%) |
> |----------------|----------------------|
> | One epoch on 3M samples |  46.1                  |
> | Two epochs on 3M samples (3M + 3M) |  48.9               |
> | One epoch on 6M samples |  55.2               |
>
>
>
> [1] Visual Instruction Tuning
>
> [2] LongVU: Spatiotemporal Adaptive Compression for Long Video-Language Understanding
>
>
> ### 2. **Clarifying Hyperparameters**
>
> The reason we use $\beta$ (set to 10) to balance the navigation loss $L=\beta L_{nav} + L_{qa}$ is that $L_{qa}$ is a cross-entropy loss, which is typically larger than the mean square loss $L_{nav}$ (computed between the ground truth waypoints and the predicted waypoints). Specifically, we conduct experiments to investigate the influence of $\beta$ on different scales of training data in VLN-CE RxR:
>
> | $\beta$ | Training Scale |  Success Rate (%) |
> |----------------|----------------------|----------------------|
> | 1  | 1M | 21.1                  |
> | 1  | 3M | 44.0                  |
> | 10 | 1M | 34.7                  |
> | 10 | 3M | 46.1                  |
>
>
> We find that $\beta$ is particularly important when the training scale is small, as a larger $\beta$ can accelerate convergence. However, with larger-scale datasets, the performance tends to converge even across different scaling factors. This suggests that our model is not sensitive to $\beta$ when large-scale data is used. We believe that a more adaptive approach to adjusting $\beta$ could be a promising direction for future work. This discussion is highlighted in Section 2.1.3.

---

### Official Review · Reviewer_xMYL · 2025-10-26

**Soundness:** 4
**Presentation:** 3
**Contribution:** 3
**Rating:** 8
**Confidence:** 4

**Summary:**

The paper presents NavFoM, a unified foundation model for cross-embodiment and cross-task embodied navigation. The model is designed to handle diverse camera setups (single or multi-view) and temporal contexts using two key mechanisms: Temporal–Viewpoint Indicator (TVI) tokens that encode spatiotemporal context, and Budget-Aware Temporal Sampling (BATS) that efficiently selects observation tokens under computational constraints.

Overall, this is a well-motivated and timely contribution that advances the goal of general-purpose embodied AI by unifying perception, language, and control within a single framework.

**Strengths:**

1. NavFoM is trained on a large-scale, diverse dataset encompassing multiple embodiments and navigation-related tasks. This scale enables effective transfer and cross-task generalisation.

2. The introduction of TVI tokens and the BATS sampling scheme are well-motivated, domain-specific innovations that address the unique challenges of temporal-spatial consistency and token efficiency in navigation settings.

3. By using a single backbone to handle instruction-following, object search, and driving tasks, the paper takes a meaningful step toward a generalist embodied navigation model. The paper reports competitive or state-of-the-art performance across multiple benchmarks and demonstrates successful deployment on real robots.

**Weaknesses:**

1. The real-world experiments rely on remote inference using an RTX-4090 GPU, as the model is too large to be hosted on typical on-board computing hardware. While this approach is understandable given current hardware limitations, it limits the practical deployability of the proposed system. Exploring model compression, distillation, or quantisation strategies could make NavFoM more suitable for real-time, on-board navigation scenarios. The reviewer acknowledges that such challenges are common in large foundation models but believes they warrant explicit discussion in the paper.
2.
* The performance gains reported in Table 1 are often modest relative to baselines, and the paper prioritises the success rate (SR) metric in the main text while other metrics receive minimal attention. A clearer explanation of the relative importance and interpretation of these metrics would improve transparency.
* Additionally, for metric-related discussion the readers are referred to the appendix. The appendix provides only a brief description of the metrics, mainly full-form expansions without equations or methodological details. Including formal definitions and clarifying what each metric measures would help readers better assess the quantitative improvements.
3. The paper does not explicitly discuss limitations, failure modes, or robustness issues, which is especially important for large foundation models deployed in real-world environments. Acknowledging potential failure cases (e.g., under sensor noise, unseen environments, or dynamic obstacles) and discussing safety implications would provide a more balanced and credible presentation of the work.
4. While the paper presents ablations on TVI and BATS components, the analysis remains somewhat surface-level. The interactions between these modules, their computational overhead, and their contribution to overall model complexity are not extensively studied. A more detailed examination (for example, isolating how each mechanism influences learning dynamics or token efficiency) would strengthen the methodological understanding and reproducibility of the work.

**Questions:**

1. Figure 3:Could the authors clarify whether Figure 3 is an illustration of the proposed TVI tokenization or an empirical plot generated from a specific task? If it represents actual performance data, additional details on the experimental setup and interpretation would be appreciated.
2. Fine-tuning: Have the authors evaluated NavFoM under a fine-tuning regime on at least one representative navigation task? Reporting such results would help assess how well the foundation model adapts to task-specific settings and whether it can surpass specialised policies after adaptation.
3. Clarification on data proportions in Figure 7: The description of “x% others + specific data” in Figure 7 is somewhat ambiguous. Could the authors specify the absolute scale of “other” versus “specific” data used to construct each mixture? Without this information, it is difficult to interpret the proportional relationships or judge whether performance improvements stem from data diversity or data volume.
4. Interactions between BATS and TVI: While Figure 8 presents independent ablations of the BATS and TVI components, it would be valuable to investigate their combined or hybrid variants (e.g., Linear Probability Sampling + Individually Learned Special Tokens). Have the authors explored whether these mechanisms exhibit complementary effects when jointly optimised?
5. Relative contribution of TVI and BATS: From the reported ablations, TVI appears to contribute more significantly to performance consistency: models including TVI often perform comparably well even when alternative sampling schemes replace BATS. Is this interpretation accurate, and can the authors comment on why TVI might play a more critical role in the unified framework?
6. Evaluation in Table 4: In the NavTest results, was NavFoM fine-tuned on the NavSim dataset, or are these results based on zero-shot transfer? Please clarify how the comparisons to baseline models were normalised to ensure an equitable evaluation (e.g., identical data exposure, observation viewpoints, training epochs, or evaluation protocols).
7. Latency and on-board scalability: Given that navigation tasks often require real-time control, could the authors provide quantitative information about inference latency and computational scaling? Insights into how NavFoM performs under constrained hardware or whether lightweight variants (e.g., distilled or quantised versions) are feasible would help evaluate its deployability in practical robotic systems.

---

> ### Author Response · Authors · 2025-11-21
> **Rebuttal - Part I**
>
> We are delighted that the reviewer xMYL recognizes the strong motivation behind developing cross-task and cross-embodiment navigation methods, and finds our designs innovative and effective in addressing the challenges in this field. We are encouraged that the reviewer thinks our performance is strong on benchmark and real-world robots. We also thank the reviewer for all the constructive feedback, which has helped us strengthen the paper. We hope the following revisions and clarifications can address the reviewer's remaining concerns:
>
>
> ### 1. **Computational cost analysis and on-board deployment**
>
> We have conducted on-board deployment by quantizing (Bitsandbytes [1]) our model (7B LLM + 2B ViT, 2048 Token Budget, four-camera view setting) from 16-bit to 4-bit. We find that this technique significantly reduces the memory cost from 19.8 GB to 10.7 GB (about a 50% decrease). Additionally, we found that our method can run directly on a onboard graphics card Jetson Thor, achieving around 2Hz trajectory prediction.
>
> | Model Version | Converged Mem Cost | Converged Inference Speed |
> |----------------|----------------------|----------------------|
> | RTX 4090 Original (16-bit) | 19.8 GB               | 218 ms               |
> | RTX 4090 Quantized (4-bit) | 10.3 GB               | 248 ms               |
> | Jetson Thor Original (16-bit) | 19.1 GB               | 566 ms               |
>
>
> Moreover, to investigate the performance after quantization (from 16-bit to 4-bit), we conducted a comparison on the VLN-CE RxR val-unseen splits. We found that the quantized model only suffers from a slight performance gap (1.4% in SR and 0.8% in SPL), which is acceptable considering the reduction of about half the memory cost.
>
> | Model Version |  SR$\uparrow$(%) |  SPL$\uparrow$(%) |  NE$\downarrow$ |   nDTW$\uparrow$
> |----------------|----------------------|----------------------|----------------------|----------------------|
> | Original (16-bit LLM)     |  64.4              |  56.2   |    4.74        |65.8
> | Quantized (4-bit LLM)  |  63.0               |  55.4  |     4.76        | 64.9
>
>
> Here, we currently employ only a naive implementation of quantization. We believe the hardware limitations could be further addressed with the rapid development of on-board graphics cards and by leveraging more advanced LLM acceleration techniques (e.g., StreamingLLM [2] and SmoothQuant [3]). We have added this discussion and related experiments in Section 3.1 and Appendix E.
>
> [1] https://huggingface.co/docs/transformers/quantization/bitsandbytes
>
> [2] Efficient Streaming Language Models with Attention Sinks
>
> [3] SmoothQuant: Accurate and Efficient Post-Training Quantization for Large Language Models
>
> ### 2. **More elaboration on the metrics and corresponding explanation**
>
> We thank the reviewer for pointing this out. We have added a detailed explanation of the metrics in the Appendix B.2 to thoroughly describe them across different benchmarks. Additionally, we have provided further clarification for Table 1, especially regarding NE, SPL, and NDTW, where our method demonstrates clear superiority in both the single-camera and four-camera settings.

---

> > ### Author Response · Authors · 2025-11-21
> > **Rebuttal - Part II**
> >
> > ### 3. **Failure case analysis**
> >
> > We appreciate the suggestion and are now providing a more detailed analysis of the failure cases, covering both benchmark and real-world environments.
> >
> > `Benchmark Environments`: We analyze benchmark failure cases in VLN-CE RxR. Our findings indicate that the limited field of view (FoV) in the single-camera setup significantly affects the ability to ground visual information with instructions. When switching to a four-camera setting (360° FoV), the success rate increases from 57.4% to 64.4%. We further analyzed failure cases and summarize the major categories below:
> >
> >
> > |  Simulator Defect | Misleading Instructions | Inaccurate Action (e.g. early turn) | Incorrect Action (e.g. wrong stop) |
> > |----------|----------|----------|----------|
> > |       26%        |           25%           |       9%      |    40%     |
> >
> >
> >
> > We observe that about 51% of failures are due to dataset/simulator problems, including rendering quality and misleading instructions (e.g., ambiguous landmarks). The remaining failures stem from model capability issues (49%), such as failing to align history with instructions (e.g., performing early stops) or failing to execute sufficient turns (especially at challenging narrow corners). This indicates that future efforts should focus on improving both datasets/simulators and model capabilities.
> >
> >
> >
> >
> > `Real-world Environments`: During the real-world experiments, we find that most failure cases stem from recognizing small objects (such as bottles or books) from a long distance or understanding blurred images while the robot is moving. Additionally, extremely challenging scenarios, such as following long-horizon instructions (thousands of words) or searching for an object within a very large building (hundreds of square meters), pose critical challenges to the method. We believe that a more robust real-world approach requires collaborative efforts in both model capabilities (perception, reasoning, memory) and hardware components (camera, computational resources).
> >
> > We agree with the reviewer that an analysis of the failure cases could help readers better understand NavFoM and guide future research. Therefore, we have added this analysis in Paper Appendix F.
> >
> > ### 4. **Analysis of the respective contributions of TVI tokens and BATS**
> >
> > We thank the reviewer for the suggestion to analyze the relationship between TVI tokens and BATS. In summary, TVI tokens provide temporal information by indicating the time steps of the given frames, while the sampling strategy of BATS enables a stochastic frame distribution that also serves as a data augmentation technique. To validate the improvements, we follow the reviewer's suggestion and conduct an ablation study on hybrid variants of the sampling strategy and embedding strategy:
> >
> >
> > | Model Version  |  SR$\uparrow$(%) |  SPL$\uparrow$(%) |  NE$\downarrow$ |   nDTW$\uparrow$ |
> > |----------------|----------------------|----------------------|----------------------|----------------------|
> > | Handcraft Tokens + Uniform Sampling  |    48.0  |    38.3   |         7.42     |    50.3      |
> > | Handcraft Tokens + BATS |         53.6       |     46.1           |    6.06      |     58.0       |
> > | TVI Tokens + Uniform Sampling  |    62.4  |    54.0   |         4.90     |    63.9      |
> > | TVI Tokens + BATS |  64.4               |  56.2               |  4.74   |   65.8    |
> >
> > Nevertheless, we find that our two key techniques lead to improvements in different model variants. Notably, handcrafted tokens (as described in Figure 8, which eliminates angle and time information) result in a corresponding performance drop. This validates the effectiveness of our designs and demonstrates the synergy between them.

---

> ### Author Response · Authors · 2025-11-21
> **Rebuttal - Part III**
>
> ### 5. **Specific fine-tuning of NavFoM**
>
> Following the reviewer's suggestion, we fine-tuned NavFoM with additional tracking data (897K) and evaluated the fine-tuned model on both the tracking benchmark (EVT-Bench distracted target, fine-tuned) and the VLN-CE RxR benchmark (cross-benchmark) using a four-camera setting.
>
> | Model Version  |  Tracking (SR) |   VLN (SR)        |
> |----------------|----------------------|----------------------|
> | Cross-tasks fine-tuning |         62.0              |  64.4               |
> | Cross-tasks fine-tuning -> Tracking Fine-tuning  |    62.5        |       47.6          |
>
> We find that fine-tuning on a specific task may yield slightly better performance (0.5%), but it suffers a significant performance drop (16.8%) when deployed on other tasks. This is due to differences in action distribution (tracking has a higher stop ratio than VLN) and instructions (tracking only requires short-horizon trajectory planning, compared to the long-horizon alignment needed in VLN). Furthermore, fine-tuning on specific data can damage real-world performance due to the restricted RGB domain of tracking-only data. Therefore, we believe that a general navigation capability model requires diverse co-tuning with navigation data.
>
>
>
>
> ### 6. **More details of Figure 3, Table 4 and Figure 7**
>
> We thank the reviewer for the suggestion to add more details about our experiments. A detailed elaboration is provided below:
>
> - `Figure 3`: We performed clustering [1] directly on the end-to-end learned TVI tokens (Eq. 3) and visualized the embeddings using a color map based on viewpoint angle and time step. Specifically, for the navigation task (Eq. 3 row 1), we sampled 1,800 TVI token embeddings from combinations of 60 angles (distributed over [0, 2π]) and 30 time steps (ranging from 0 to 150). For the VQA task (Eq. 3 row 2), we sampled embeddings from 30 time steps ranging from 0 to 150. We add more details in Appendix A.3.
>
> - `Table 4`: The training data for NavFoM includes both NeuScene and Openscene datasets (NavSim data is a split of OpenScene). We mixed these autonomous driving datasets with other navigation data, without specific fine-tuning on NavSim. This is enabled by the use of TVI tokens, which allow NavFoM to handle different camera setups and support unified training and evaluation. Our evaluation strategy is consistent with existing baseline [2] to ensure a fair comparison. We highlight this in Appendix B.1.
>
> - `Figure 7`: Following the reviewer's suggestions, we have provided the absolute scale of each component in the training data. The corresponding performance metrics are listed below in the format (other data / task-specific data / performance).
>
> | Task |  0% Others + Specific task   | 50% Others + Specific task   |  100% Data  |
> |----------------|----------------|----------------------|----------------------|
> | VLN (SR)|  0M/3.37M/57.5                  |      2.32M/3.37M/61.7           |       4.65M/3.37M/64.4           |
> | Searching (SR)|    0M/1.02M/10.3               |  3.5M/1.02M/39.1            |          7M/1.02M/45.2      |
> | Tracking (SR)|  0M/897K/12.6                  |  3.56M/897K/39.0               |        7.1M/897K/62.0         |
> | Driving (PDMS)|  0M/681K/79.4                 |   3.66M/681K/83.1              |       7.33M/681K/84.3       |
>
>
> [1] Umap: Uniform manifold approximation and projection for dimension reduction
>
> [2] DiffusionDrive: Truncated Diffusion Model for End-to-End Autonomous Driving

---

> > ### Comment · Reviewer_xMYL · 2025-11-23
> >
> > My concerns about the paper have been adequately addressed, so I am happy to maintain my acceptance recommendation as is. I appreciate the authors’ efforts in incorporating the Failure Case Analysis into the revised manuscript, as well as the additional results and ablation studies they have conducted. I believe these additions genuinely help to clarify and strengthen the paper's interesting findings. One small suggestion would be to reference the Failure Case Analysis section within the main text (maybe around section 3.2 or 3.3) to enhance its visibility and readability.
> >
> > I have a few questions that might be of interest to future practitioners and researchers:
> >
> > 1. I found the new quantization experiments related to computational cost very interesting. The fact that quantizing from 16-bit to 4-bit results in only minimal performance degradation is quite remarkable. Do you think techniques such as Quantization-Aware Training or Distillation (for eg., using a Teacher-Student approach) could extend this work to enable real-time control on actual hardware for robotics? What insights and techniques can the authors provide from their experience?
> >
> > 2. Regarding the "Specific fine-tuning of NavFoM" experiment: Could the authors provide more details on this? It makes sense that task-specific fine-tuning could lead to a performance drop when the model is deployed on other tasks. However, the observed drop seems quite significant compared to the relatively small (0.5%) performance gain on the specific task itself. This could also depend on several factors, including fine-tuning steps, learning rate, etc. Given this, I have some follow-up questions:
> >
> > (a) Does this suggest that the proposed foundation model is generally not amenable to task-specific fine-tuning?
> >
> > (b) What level of performance should one expect from a specialist policy trained on that specific task?
> >
> > (c) Additionally, what level of computational resources should future researchers expect to have available if they wish to fine-tune the generalist model for their specific navigation task or problem domain?

---

> > > ### Author Response · Authors · 2025-11-25
> > > **Response to Reviewer (Part I)**
> > >
> > > We are glad that our rebuttal helps to address your concerns. We have also followed the suggestion and added the reference to the Failure Case Analysis in Section 3.2. The remaining questions are answered below:
> > >
> > > ### **Performance after Quantization**
> > >
> > > The performance gap between 16-bit and 4-bit quantization under certain tasks being small has been widely reported in both LLMs [1] and VLAs [2]. In particular, for the manipulation VLA method, OpenVLA [2], results show (OpenVLA Table 2) that 4-bit quantization achieves a bridge success rate of `71.9 ± 4.7%`, which is a very strong result compared to the 16-bit performance of `71.3 ± 4.8%` (even slightly outperforming it). We believe the reason behind this is that online robot tasks, such as manipulation and navigation, can mitigate the negative impact of 4-bit quantization through their closed-loop nature, which continuously updates action predictions and thereby reduces the accumulation of inaccuracy caused by quantization.
> > >
> > > Regarding deployment memory cost, techniques such as SmoothQuant [3] and quantization-aware training could significantly reduce memory usage while maintaining strong performance. For inference speed, there are also existing advanced techniques such as LLM streaming [4] (which is suitable for processing online captured video in robotic tasks) and the Speculative Decoding strategy [5]. These methods have demonstrated significant inference speed improvements in complicated tasks [6]. We have added this discussion to the Appendix F.
> > >
> > >
> > > [1] QLoRA: Efficient Finetuning of Quantized LLMs
> > >
> > > [2] OpenVLA: Open-Vocabulary Large Action Models for Robotic Manipulation
> > >
> > > [3] Smoothquant: Accurate and efficient post-training quantization for large language models.
> > >
> > > [4] Efficient streaming language models with attention sinks.
> > >
> > > [5] Fast inference from transformers via speculative decoding. In International Conference on Machine Learning
> > >
> > > [6] Lumine: An Open Recipe for Building Generalist Agents in 3D Open Worlds

---

> ### Author Response · Authors · 2025-11-25
> **Response to Reviewer (Part II)**
>
> ### **Fine-tuning Performance**
>
> For training details, the tracking fine-tuning (Stage 2) is conducted with the same batch size and learning rate schedule (cosine) as the cross-task fine-tuning (Stage 1), except that the learning rate is reduced from 2e-5 to 5e-6 (a strategy commonly adopted in fine-tuning VLMs and VLAs methods [1,2]). For the training samples, Stage 1 fine-tuning follows the same setup as described in NavFoM (Section 2.2), which also includes tracking data. In Stage 2, the model is fine-tuned for one additional epoch using only tracking data. Note that both Stage 1 and Stage 2 are referred to as fine-tuning since NavFoM builds upon the pre-trained weights of the LLM and ViTs.
>
> Therefore, if the model already knows how to track, the specific fine-tuning stage may only yield a marginal gain. In this case, we further provide an experiment in which `stage 1 fine-tuning is performed without tracking data, and stage 2 fine-tuning uses tracking data only` (requiring approximately 16 H100 GPUs for 8 hours of training, totaling 128 GPU hours). The results are shown below:
>
> | Model Version  |  Tracking (SR) |   VLN (SR)        |
> |----------------|----------------------|----------------------|
> | Stage 1 fine-tuning (all data) |         62.0              |  64.4               |
> | Stage 1 fine-tuning (all data) -> Tracking fine-tuning (tracking only)  |    62.5        |       47.6          |
> | **Stage 1 fine-tuning (except tracking data)**  |    14.1       |       62.1          |
> | **Stage 1 fine-tuning (except tracking data) -> Tracking fine-tuning (tracking only)**  |    59.7       |       44.3          |
>
> We find that if the Stage 1 fine-tuning does not include tracking data, there is a significant performance gain (`from 14.1% to 59.7%`) after fine-tuning on tracking data only. However, if the pre-training stage already includes tracking data, the fine-tuning stage brings only marginal gains (from 62.0% to 62.5%), along with a noticeable performance drop on the VLN task (from 64.4% to 47.6%). We believe that fine-tuning can yield significant improvements (row3 -> row 4) only when the model has not yet adequately developed the specific capability in the previous training stage (row 3). In contrast, if the model has already mastered the capability (row 1), further fine-tuning (without incorporating higher-quality or more diverse data) may bring only limited gains or even harm generalization (row 2). These findings are consistent with the literature on LLM-based training [1]. Furthermore, several VLA studies, such as Pi0 [2], have reported that excessive fine-tuning on specific data can degrade performance on certain tasks (e.g., 10 hours of fine-tuning on Paper Replacement and Towel Folding leads to worse results than 5 hours). We also want to highlight the multi-task synergy in LLM-based training (as a highly sample-efficient method), where cross-task fine-tuning could lead to synergistic effects among different tasks, which could improve overall performance (row 4 vs. row 1) [3].
>
> Since the cross-task fine-tuning versus specific-task fine-tuning debate is an ongoing topic that involves many factors, such as data quality, training strategy, and model capabilities, our empirical conclusion is that, if computational resources and data scale allow, training on diverse data in a cross-task manner is preferred, as it can bring better generalization ability across overall tasks. However, if computational resources are limited or only a limited amount of high-quality specific-task data is available, we encourage performing specific-task fine-tuning (which requires 128 H100 GPU hours for 897K samples. This could vary slightly if the navigation horizon is significantly smaller or larger).
>
> If the reviewer considers it necessary, we would be happy to conduct specific-task fine-tuning on other specific tasks (VLN, Searching, and Driving) and include the results in the camera-ready version accordingly. (Due to the extensive computational costs, this may not be finished within the rebuttal period.)
>
>
> [1] Language Models are Few-Shot Learners
>
> [2] Pi0: A Vision-Language-Action Flow Model for General Robot Control
>
> [3] Uni-NaVid: A Video-based Vision-Language-Action Model for Unifying Embodied Navigation Tasks

---

> > ### Comment · Reviewer_xMYL · 2025-11-25
> >
> > Thank you for the detailed clarification. I appreciate the authors' willingness to conduct additional fine-tuning on other tasks, though I understand the practical constraints within the current timeframe.
> >
> > Overall, I think this paper makes a solid contribution, and I’m sticking with my original acceptance recommendation. Most of my follow-up points were broader questions around foundation model usage and fine-tuning practices. I asked them with the hope of encouraging some helpful discussion and documentation, especially here on openreview. It could help future researchers and adopters better understand the practical challenges and set more realistic expectations when it comes to fine-tuning large models. Thanks again for the productive discussion!

---

### Official Review · Reviewer_VJ2q · 2025-10-29

**Soundness:** 3
**Presentation:** 3
**Contribution:** 3
**Rating:** 8
**Confidence:** 2

**Summary:**

NavFoM adopts a unified architecture that can handle multimodal inputs from different camera configurations and time spans, making it suitable for tasks across various embodiment types, such as quadruped robots, drones, wheeled robots, and vehicles. These tasks include vision-language navigation, object search, object tracking, and autonomous driving. To address multi-view and temporal contexts, the model introduces Temporal-View Indication (TVI) tokens, which help the model understand and adapt to different camera perspectives and time steps. Through an innovative Budget-Aware Temporal Sampling (BATS) strategy, NavFoM efficiently manages navigation history information with limited computational resources, balancing performance and inference speed. Moreover, NavFoM achieves state-of-the-art performance on multiple public benchmarks and demonstrates strong generalization capabilities across various embodiments and tasks, particularly excelling without the need for task-specific fine-tuning. Through practical testing on platforms such as humanoid robots, quadruped robots, drones, and wheeled robots, the paper validates the model's potential for real-world applications, showing that NavFoM can effectively handle complex navigation tasks with broad practical applicability.

**Strengths:**

1.Unified Architecture: The model processes multimodal navigation inputs from different camera configurations and task horizons. It uses temporal-viewpoint indicator tokens (TVI tokens) to accommodate diverse camera setups and time contexts in navigation tasks. This allows the model to handle various visual inputs dynamically.

2.Budget-Aware Temporal Sampling (BATS): This method addresses real-world deployment challenges, such as memory and inference speed, by dynamically sampling navigation history based on a token budget. This ensures the model balances performance with resource constraints.

3.Generalization Across Tasks and Embodiments: NavFoM achieves state-of-the-art (SOTA) performance across seven public benchmarks and in real-world experiments without requiring task-specific fine-tuning. The model demonstrates significant improvements, particularly in multi-camera settings, and shows robust performance in diverse environments such as robotics, drones, and autonomous vehicles.

**Weaknesses:**

A large language model based approach might fail to handle highly dynamic environments.

**Questions:**

1. The output of the model is two-dimensional. How can we control the drone to navigate in a three-dimensional space?
2. Will using the latest model, such as Qwen3 VL, lead to performance improvement?

---

> ### Author Response · Authors · 2025-11-21
> **Rebuttal**
>
> We are delighted that the reviewer VJ2q  finds our unified architecture innovative and well-motivated, recognizes the technical strength of our approach in handling diverse multimodal inputs, and considers our experimental results strong and convincing, with state-of-the-art generalization across tasks and embodiments. We also thank the reviewer for the constructive feedback, which has helped us strengthen the paper through revisions. We hope the following revisions and clarifications address the reviewer's remaining concerns.
>
> ### 1. **Performance in Highly Dynamic Environments**
>
> This is a very good point that highly dynamic environments, such as tracking a running person within crowded environments, require consistent and fast inference speed. This is particularly challenging for large model-based methods, as increased token length and large-scale inference parameters lead to longer inference times.
>
>
> We propose the BATS sampling strategy to alleviate the inference speed challenges of large models. BATS enables NavFoM to maintain a compact token sequence without exceeding the token budget, which leads to consistent and fast inference speed. Below, we list the inference speed benchmarks across different token budgets in a single-camera setting:
>
> | Model Version |  Converged Inference Speed |
> |----------------|----------------------|
> | 2048 token budget |  192ms |
> | 1024 token budget |  126ms |
> | 512 token budget  |  107ms |
>
>
> Our method can achieve an inference speed of about 8 Hz with 1k visual tokens (over 200 steps), which is close to the requirements of some applications (e.g., approximately 10 Hz for autonomous driving). We believe this inference speed could be further improved with more advanced LLM acceleration techniques [1,2], and we plan to investigate them in the future. We have added a corresponding discussion in Paper Appendix F.
>
> [1] Efficient Streaming Language Models with Attention Sinks
>
> [2] SmoothQuant: Accurate and Efficient Post-Training Quantization for Large Language Models
>
> ### 2. **Clarification on the action dimension**
>
>
> NavFoM employs a unified action planning head that predicts four-dimensional waypoints (a unified upper-bound dimension for different embodiments), comprising x, y, z, and $\theta$. To accommodate varying task dimensions across different navigation scenarios (e.g., three-dimensional (x, y, $\theta$) for ground robots and four-dimensional (x, y, z, $\theta$) for UAVs), we normalize trajectories to the range [-1, 1] and apply a `dimension mask` (L287) to mask unused dimensions during both training and inference (Eq. 6). Note that our approach can be further enhanced with advanced action planning techniques [1,2], and we have highlighted this discussion in the revised manuscript (Section 2.1.3).
>
>
> [1] Diffusion Policy Visuomotor Policy Learning via Action Diffusion
>
> [2] TrackVLA: Embodied Visual Tracking in the Wild
>
> ### 3. **Performance on stronger base model**
>
> We thank the reviewer for the suggestion of ablating the effect of the base model. Due to the limited time for rebuttal, it is challenging to appropriately adjust Qwen3-VL and fully train it on 12M data, which would require re-generating the offline visual feature cache. Nevertheless, we provide experimental results where we changed the base language model from (Vicuna-7B + EVA-CLIP) to (Qwen2.5-7B + SigLiP + Dinov2) on the VLN-CE RxR four-camera setting.
>
> | Base Model  |  SR$\uparrow$(%) |  SPL$\uparrow$(%) |  NE$\downarrow$ |   nDTW$\uparrow$ |
> |----------------|----------------------|----------------------|----------------------|----------------------|
> |  Vicuna-7B + EVA-CLIP  |  59.2              |  47.1   |    5.42        |58.0
> |  Qwen2.5-7B + SigLiP + Dinov2   |  64.4               |  55.4  |     4.76        | 64.9
>
>
> From the results, we find that switching to a stronger base model can significantly improve performance. We have to mention that this experiment is not entirely fair, as we did not have sufficient time to ensure the training configurations were fully aligned. However, we believe it is widely acknowledged that a better base model generally leads to better performance [1]. We will follow the reviewer's suggestion and include the results of using Qwen3-VL as the base model in the camera-ready version.
>
> [1] Towards Generalist Robot Policies: What Matters in Building Vision-Language-Action Models

---

### Official Review · Reviewer_E4q1 · 2025-10-30

**Soundness:** 4
**Presentation:** 4
**Contribution:** 4
**Rating:** 10
**Confidence:** 4

**Summary:**

The primary contribution of this research is the design and training of a unified navigation foundation model (NavFoM). This model was trained on a dataset comprising eight million navigation samples, encompassing diverse robotic platforms and varied navigation tasks. To handle the varying camera configurations across platforms and the temporal span of tasks, NavFoM incorporates identifier tokens to embed both view information and temporal context. Furthermore, to accommodate computational resource constraints in practical deployments, the model employs a sampling strategy that dynamically adjusts observation tokens within a limited token length budget. Evaluation results demonstrate that NavFoM achieves leading or highly competitive performance across seven public benchmarks without requiring task-specific fine-tuning. Real-world robotic experiments further validate its robust generalization capabilities and practical applicability.

**Strengths:**

1. The paper introduces the core concept of a “cross-vehicle, cross-task navigation foundation model,” demonstrating high innovation and foresight. It addresses the limitation in current embodied AI navigation research where models are often confined to specific vehicles or tasks, pointing toward a promising direction for constructing universal navigation agents.
2.The paper's technical solution for unifying multi-vehicle, multi-task inputs—using identifier tokens to embed camera view information and task temporal context—is remarkably clever and direct. The dynamic token sampling strategy also demonstrates deep consideration for practical deployment constraints.
3.Extensive testing across seven public benchmarks and real-world scenarios covering diverse robots and tasks provides robust validation of NavFoModel's effectiveness and generalization capabilities, lending strong credibility to the results.
4.The research does not confine itself to idealized laboratory settings but explicitly addresses and tackles challenges in real-world deployments, such as computational efficiency and adaptability to dynamic environments, thereby enhancing the practical value of the work.
5.The author provided detailed implementation specifics and stated that the source code would be open-sourced.

**Weaknesses:**

1.The paper would be significantly strengthened by adding a dedicated analysis of the model's failure modes (e.g., in long-horizon planning or perceptually challenging environments), which is vital for defining the approach's boundaries and guiding future research.​​
2.The paper emphasizes the model's practicality and proposes the BATS strategy to optimize computational overhead, but lacks detailed, quantitative analysis of deployment costs.

**Questions:**

see weekness

---

> ### Author Response · Authors · 2025-11-21
> **Rebuttal - Part I**
>
> We are delighted that the reviewer E4q1 finds our method innovative in embodied navigation, considers the paper technically sound and well-organized, and agrees that our experimental results are strong and convincing. We also thank the reviewer for the constructive feedback, which has helped us significantly strengthen the manuscript. We hope the following revisions and clarifications address the reviewer’s remaining concerns.
>
> ### 1. **Failure case analysis**
>
> We appreciate the suggestion and are now providing a more detailed analysis of the failure cases, covering both benchmark and real-world environments.
>
> `Benchmark Environments`: We analyze benchmark failure cases in VLN-CE RxR. Our findings indicate that the limited field of view (FoV) in the single-camera setup significantly affects the ability to ground visual information with instructions. When switching to a four-camera setting (360° FoV), the success rate increases from 57.4% to 64.4%. We further analyzed failure cases and summarize the major categories below:
>
>
> |  Simulator Defect | Misleading Instructions | Inaccurate Action (e.g. early turn) | Incorrect Action (e.g. wrong stop) |
> |----------|----------|----------|----------|
> |       26%        |           25%           |       9%      |    40%     |
>
>
>
> We observe that about 51% of failures are due to dataset/simulator problems, including rendering quality and misleading instructions (e.g., ambiguous landmarks). The remaining failures stem from model capability issues (49%), such as failing to align history with instructions (e.g., performing early stops) or failing to execute sufficient turns (especially at challenging narrow corners). This indicates that future efforts should focus on improving both datasets/simulators and model capabilities.
>
>
>
>
> `Real-world Environments`: During the real-world experiments, we find that most failure cases stem from recognizing small objects (such as bottles or books) from a long distance or understanding blurred images while the robot is moving. Additionally, extremely challenging scenarios, such as following long-horizon instructions (thousands of words) or searching for an object within a very large building (hundreds of square meters), pose critical challenges to the method. We believe that a more robust real-world approach requires collaborative efforts in both model capabilities (perception, reasoning, memory) and hardware components (camera, computational resources).
>
> We agree with the reviewer that an analysis of the failure cases could help readers better understand NavFoM and guide future research. Therefore, we have added this analysis in Paper Appendix F.

---

> ### Author Response · Authors · 2025-11-21
> **Rebuttal - Part II**
>
> ### 2. **Deployment costs**
>
> We thank the reviewer for the suggestion, and we have conducted additional experiments on deployment costs. Specifically, we provide the original costs (16-bit) and quantized version (4-bit) of our model (`7B LLM + 2B ViT, 2048 Token Budget, four-camera view`) on VLN-CE RxR in the table below:
>
>
>
> | Model Version | Converged Mem Cost | Converged Inference Speed |  SR$\uparrow$(%) |  SPL$\uparrow$(%) |  NE$\downarrow$ |   nDTW$\uparrow$
> |----------------|----------------------|----------------------|----------------------|----------------------|----------------------|----------------------|
> | Original (16-bit LLM) | 19.8 GB               | 218 ms               |  64.4              |  56.2   |    4.74        |65.8
> | Quantized (4-bit LLM) | 10.3 GB               | 248 ms               |  63.0               |  55.4  |     4.76        | 64.9
>
>
> We find that our quantized models (4-bit via Bitsandbytes [1]) significantly reduce deployment costs (using only 53.9% of the original memory) while maintaining comparable navigation performance. Our approach can also be deployed on the latest onboard GPU, the NVIDIA Jetson Thor, using 16-bit precision, achieving an average inference speed of 566 ms per trajectory prediction. Here, we currently employ only a naive implementation of quantization. This can be further improved with more advanced acceleration techniques (e.g., StreamingLLM [2] and SmoothQuant [3]). In summary, we believe that with the rapid development of graphics hardware and acceleration methods, fast and convenient deployment of large model-based approaches will become a promising direction.
>
> We add this discussion and experiment in Sec 3.1 and Appendix E.
>
> [1] https://huggingface.co/docs/transformers/quantization/bitsandbytes
>
> [2] Efficient Streaming Language Models with Attention Sinks
>
> [3] SmoothQuant: Accurate and Efficient Post-Training Quantization for Large Language Models

---

> > ### Comment · Reviewer_E4q1 · 2025-11-27
> > **Reply after rebuttal**
> >
> > The author's response addressed my concerns and revealed exciting potential applications. Additionally, reviewing other replies deepened my understanding of the author's experiments on training details and hyperparameters.

---

### Author Response · Authors · 2025-12-02
**Global Response**

We sincerely thank all reviewers for their valuable comments and suggestions. We appreciate that the reviewers recognized our work as well-motivated (`E4q1, VJ2q, xMYL`), featuring novel techniques (`E4q1, VJ2q, xMYL, iLxn`), and demonstrating strong performance (`E4q1, VJ2q, xMYL, iLxn`).

We summarize our responses to the reviewers' comments as follows:

- We provide detailed experiments on the deployment cost of NavFoM, including memory usage and inference speed after quantization (`saving about 50% of the memory`) and for onboard deployment (`running at 2 Hz on Jetson Thor`).
- We provide failure case analysis across different tasks, along with potential discussions to improve them in the future.
- We provide additional ablation studies and analyses, including the respective contributions of TVI tokens and BATS, as well as the specific fine-tuning of NavFoM.
- We provide more details about our experiments, including the metrics used in Figure 3, Table 4, and Figure 7.

We appreciate the time invested by the reviewers, ACs, and SACs in helping us improve our paper. We are glad that our responses have addressed the concerns of `E4q1` and `xMYL`, who replied before the discussion ended. We are committed to incorporating the above experiments and adding further discussion as suggested by the reviewers in the camera-ready version.

---

### Meta-Review · Area_Chair_geJC · 2026-01-09

**Summary:**

This paper proposes NavFoM, a unified navigation foundation model that innovatively integrates cross-embodiment and cross-task inputs using Temporal-Viewpoint Indicator tokens and a Budget-Aware Temporal Sampling strategy. Reviewers recognized its contribution, elegant technical solutions, and comprehensive evaluation across seven benchmarks and real-world tests, marking a significant step towards a generalist navigation agent. While recommending acceptance, they highlighted key areas for improvement: a lack of detailed analysis on failure modes, robustness, and deployment costs; insufficient discussion on real-time on-board feasibility and model limitations; and requests for clarification on fine-tuning results, data composition, and component interactions.

**Reviewer Concerns:**

All reviewer concerns are well addressed by the author's rebuttal and the feedback from reviewers. Area Chair (AC) is in agreement with the positive assessment provided by the four reviewers.

**Reviewer Scores:**

All reviewers have already reached a consensus to accept the paper. The AC has thoroughly reviewed the comments and rebuttal and agrees with the overall evaluation. Consequently, the AC recommends accepting this paper for publication.

---

### Decision · Program_Chairs · 2026-01-26

Accept (Poster)